# Upcycling Polystyrene

**DOI:** 10.3390/polym14225010

**Published:** 2022-11-18

**Authors:** Jaworski C. Capricho, Krishnamurthy Prasad, Nishar Hameed, Mostafa Nikzad, Nisa Salim

**Affiliations:** School of Engineering, Swinburne University of Technology, Hawthorn, Melbourne, VIC 3122, Australia

**Keywords:** polystyrene, recycling, waste plastics, plastic pollution, environmental assessment, sustainability

## Abstract

Several environmental and techno-economic assessments highlighted the advantage of placing polystyrene-based materials in a circular loop, from production to waste generation to product refabrication, either following the mechanical or thermochemical routes. This review provides an assortment of promising approaches to solving the dilemma of polystyrene waste. With a focus on upcycling technologies available in the last five years, the review first gives an overview of polystyrene, its chemistry, types, forms, and varied applications. This work presents all the stages that involve polystyrene’s cycle of life and the properties that make this product, in mixtures with other polymers, command a demand on the market. The features and mechanical performance of the studied materials with their associated images give an idea of the influence of recycling on the structure. Notably, technological assessments of elucidated approaches are also provided. No single approach can be mentioned as effective per se; hybrid technologies appear to possess the highest potential. Finally, this review correlates the amenability of these polystyrene upcycling methodologies to frontier technologies relating to 3D printing, human space habitation, flow chemistry, vertical farming, and green hydrogen, which may be less intuitive to many.

## 1. Introduction

Badische Anilin und Soda Fabrik (BASF) introduced the first commercial-grade polystyrene (PS) in the 1930s. Since then, PS has captured its niche market owing to its cost-effectiveness, excellent processability, low density, clear appearance, dimensional stability, and amenability to be sterilized by radiation. Today, the applications of PS are diverse, and it has even managed to outcompete other plastics in various industries. Unfortunately, as with other plastics in the market, the utilization of PS was aimed at single-use. The disposable and lightweight nature of different PS-derived materials led them to accumulate in the environment as wastes with no facile mechanism for degradability.

The impediment to recycling PS is like those of many plastics. However, it is compounded by the negative scrap value of expanded polystyrene (EPS), the preponderant PS-derived litter ordinarily available as food packaging material. In contrast, recycled plastic fractions giving the highest profit are rigid PS packaging (2021 price levels), with a 14% internal rate of return [1]. This disparity is due to the product’s price and yield, which determines recycling’s economic incentives. Moreover, visual segregation of PS from the mixed waste stream is challenging, and the presence of other specific contaminants could significantly affect the recovery [2]. Plastics based primarily on high-impact polystyrene (HIPS) and acrylonitrile–butadiene–styrene (ABS) constitute the lion’s share of waste electrical and electronic equipment (WEEE), thus garnering increased interest in recycling these products [3]. However, the intrinsic immiscibility of the polymer’s components and the varied nature of the sourced polymers remain a challenge in advancing the recycling of WEEE as sources of next-generation polymeric materials. The plastic industry associated with EPS is now cognizant of the global necessity to recycle post-consumer EPS material. Both the private sector and academia commissioned a study [4], aptly illustrated in Figure 1, that recognized the viability of the mechanical recycling of EPS as a potential route to a circular economy model.

Today, in the age of accelerating industrialization and urbanization, our modern society’s solid waste management programs have considered the gradual closing of the circular loop in terms of sustainably recycling and reusing polymers. The main problem in utilizing post-consumer polymers over and over is the consequent degradation of their mechanical strength, often regenerating polymers of low value. This form of recycling is known as downcycling. In contrast, enhancing the mechanical strength of recycled materials or the regeneration of high-value polymers or chemicals in recycling is called upcycling [5].

This review details the various upscaling methods that have been employed in the recycling of PS-based post-consumer products. An overview of the PS as a polymer, its chemistry, and its manufacture with its myriad applications are introduced before tackling the details of recycling PS-derived materials. Whenever possible, environmental or techno-economic analyses of the discussed recycling methods are included in this review.

### 1.1. Polystyrene

Styrene is a petroleum-derived, liquid, aromatic hydrocarbon, and is the precursor monomer of polystyrene (PS). PS is a thermoplastic that is usually solid at room temperature, melts at an elevated temperature during molding or extrusion, and then solidifies. As a thermoplastic, joining or welding PS is feasible, limited only by the brittleness of the employed PS grades. The radio and ultrasonic frequency welding technique or solvent bonding can be utilized with some PS grades [6,7]. As a film, PS possesses high tensile strength and excellent transparency, but with poor gas, moisture, or vapor barrier properties. However, this low gas barrier property enables the fabrication of ‘breathable’ PS films. PS is generally brittle and exhibits less chemical resistance to organic solvents from aliphatic, aromatic, and chlorinated varieties, including acids and bases, ketones, and cyclic ethers.

Nevertheless, in dilute aqueous solutions of acids or bases and aliphatic alcohols of high molecular weight, PS is moderately resistant. On the other hand, PS exhibits chemical resistance to ethylene oxide (EO), low molecular weight alcohols, and oxidizing and disinfecting agents such as bleach. In fact, EO has been used to sterilize PS-based materials, with no significant effects on physical properties upon exposure [8]. Sterilization using steam or an autoclave is not employed with PS due to the low heat distortion temperature. Exposure to gamma and e-beam radiation is also commonly employed in sterilizing medical-grade PS-based materials without a significant compromise on the properties of PS [9]. This stability is due to the high aromatic content of PS, wherein the electronic clouds of the phenyl moieties can neutralize the reactive free radicals generated on exposure [10]. Several grades of commercially biocompatible PS copolymers are available in the market, although PS is not normally employed in applications that necessitate biocompatibility.

#### 1.1.1. Chemistry and Synthesis

The polymerization of the styrene monomer via the generation of free radicals is the main route in the synthesis of polystyrene. This free-radical polymerization employs free-radical initiators activated by thermal or radiation sources. An example of this initiator is benzoyl peroxide in Figure 2, where it is mixed with the styrene monomer with or without diluents and heated at ca. 120 °C. After quenching the reaction, the high molecular weight PS is obtained. The unreacted monomer and remaining diluent are removed using vacuum extraction.

PS can be synthesized in three different forms depending on its tacticity (Figure 3): atactic, syndiotactic PS (sPs), and isotactic PS [11]. Tacticity, in this case, refers to the spatial arrangement or order of the phenyl groups within the polymeric chain, and this arrangement influences the polymer’s properties. Commercially, atactic PS, a polystyrene having randomly distributed phenyl groups on either side of the polymeric chain, is preponderantly available. At the same time, the sPS was initially jointly introduced in Japan by the Idemitsu Petrochemical Company and Dow in 1988. The phenyl groups in sPs are in alternating positions on the polymeric chain, resulting in the semi-crystallinity of PS. The melting point of this engineering polymer, sPS, lies at ca. 270 °C and is synthesized through a continuous polymerization process similar to polyolefins that employ metallocene single-site catalysts. Although sPS is brittle, it exhibits high chemical resistance and possesses a very low dielectric constant. Glass reinforcement or blending with other polymers are a few ways to circumvent the brittleness issue of sPS. Moreover, the ease of processing and high flow of sPS are suitable material characteristics required for thin-wall applications.

#### 1.1.2. Polystyrene Grades and Applications

Polystyrene generally exists in two chemical grades: crystal polystyrene and HIPS. The crystal PS or general-purpose PS is mostly synthesized via the bulk polymerization technique and considered unmodified polystyrene. Crystal PS-derived resins are glassy and very transparent, supplied principally as granules. Physically, PS can be in solid, foam, or expanded (EPS) forms. The general-purpose or crystal polystyrene, although brittle, is the thermoplastic of choice for PS injection molding and extrusion processes. Out of the extrusion process come many packaging materials and extruded trays. The injection molding of crystal PS yields several medical components and diagnostic labware, which are not limited to syringe hubs, petri dishes, canisters, and trays for tissue culture. Some office accessories and dinnerware such as cups or tumblers, cutleries, and other plastic housewares are manufactured by the injection molding process. Foamed sheets out of extrudable crystal grade PS yield most of the commonly employed materials for meat packaging and the fast-food industry sectors. In contrast, various insulation materials for the building industry come out of foamed boards. The general-purpose PS are characteristically hard and rigid, exhibit high gloss, good electric/dielectric properties, and dimensional stability, with low water absorption and limited chemical resistance, but possess excellent resistance to γ-radiation. The processability of crystal grade PS is excellent; however, it is prone to stress cracking in specific environmental conditions. In comparison, HIPS is tougher, having improved impact resistance, and thus is mostly utilized in thermoforming. HIPS is characteristically translucent to opaque in appearance, exhibits low gloss, and has reduced electrical properties and fair dimensional stability, with reduced water absorption and resistance to chemicals, but fair resistance to γ-radiation. HIPS also exhibits excellent processability and is less prone to cracking if stressed in specific environments. Thermoformed medical products that are HIPS-derived include epidural, catheter, and heart pump trays, suction canisters, syringe hubs, and respiratory care equipment. Both chemical forms of PS compete with polypropylene (PP), poly(vinyl chloride) (PVC), or acrylic-based materials, especially for a lot of respiratory and medical products.


**High-impact polystyrene (HIPS):**


HIPS is PS with a bright white appearance and is derivatized with rubber-like moieties. During styrene polymerization, the rubber-like polybutadiene is incorporated into the growing polystyrene chains. The in situ grafting and partial cross-linking of the butadiene moieties leads to better adhesion between the rubber-like particles and the PS matrix, resulting in toughness improvement. In turn, this reaction significantly affects the final polymer’s properties. HIPS resins have characteristically good dimensional stability, rigidity, and impact strength, and are easily processable. In 3D printing, the HIPS has a wide print temperature range of ca. 210–250 °C.

The elastomeric content of high-impact grade PS ranges from 6–12%, while medium-impact grades contain about 2 to 5% elastomers. HIPS-derived PS are common commodities in the food packaging or consumer product business in the form of yogurt or dairy containers, drinking bottles, cups or tumblers, lids, and dinnerware, where impact resistance is essential. HIPS is also a component of many consumer electronics, refrigerator linings, and high-quality office appliances, and is used in the manufacturing of toys, audio-visual equipment, and medical devices. HIPS is generally biocompatible and has been used long-term in high-impact prosthetic devices with no adverse effects in contact with human tissues. Several 3D-printed surgical or dental instruments have been designed using HIPS.


**Oriented polystyrene (OPS)**


The crystal polystyrene grade remains brittle except when it is biaxially oriented, becoming flexible and durable. This stretching of the PS sheet in the transverse direction yields oriented polystyrene (OPS), resulting in the toughening of the polymer sheet. Culinary baking and delicatessen trays are fabricated from extruded OPS.


**Expanded polystyrene (EPS)**


Expanded polystyrene (EPS) is a plastic consisting of a stiffened cellular structure resulting from molded spheres or little pearls of expandable polystyrene, generally a mixture of approximately 5% PS polymer and 95% air [12]. Blowing agents like pentane or chlorofluorocarbons are needed to generate air space. Because the entrapped air fills the voids as the polymer stiffens, the material becomes lightweight with low thermal conductivity. Among the rigid insulation foams, EPS has the lowest average R-value and is the most cost-effective. For these reasons, EPS is aptly used as insulation in extrudable foamed board forms such as insulating concrete foams (ICFs) and structural insulation panels (SIPs) in building construction, and as patterned sheeting in other engineering or architectural applications, although it is fragile to work with. EPS absorbs water; however, it is rated for ground contact and can be rendered as an insect-repellent. EPS is semipermeable, and unable to create a vapor barrier. 


**Styrenic copolymers**


Styrenic copolymers are PS derivatives that are structurally similar to HIPS. The chemical derivatization of PS has been shown to improve its rigidity, chemical, and scratch resistance, among others. One of these styrenic copolymers is the Acrylonitrile-Butadiene-Styrene (ABS) Copolymer, mostly obtained from emulsion polymerization. In the synthesis of ABS, submicron rubber-like particles are initially produced from an emulsified butadiene before grafting styrene-acrylonitrile copolymer on the shell of rubber particles, which eventually leads to a grafted rubber concentrate (GRC). GRC is then finally compounded with styrene-acrylonitrile copolymers (SAN) as a toughening approach in specific applications. ABS can also be obtained by a mass process like HIPS. However, in comparison with HIPS, ABS polymers are tougher and more chemically resistant but are more challenging to process. ABS-based polymers are utilized in the interior trim of automotive components, luggage, household pipes, housing for power tools and vacuum cleaners, and in the fabrication of plastic toys.

Another styrenic copolymer is SAN. SAN is synthesized by copolymerizing the acrylonitrile (AN) monomer at 15 to 30% by weight with the styrene monomer (SM) [13]. SAN-derived materials are for applications necessitating extra attributes such as toughness, excellent chemical resistance, and high heat distortion temperature. These attributes are highly considered in the fabrication of toys, shower doors, and bottles or containers used in the cosmetic industry. SAN resins are a major component in the compounding of ABS resins by emulsion polymerization.

Other styrenic copolymers also exist, including maleic anhydride copolymerized with styrene and transparent impact polystyrene (TIPS). Styrene and maleic anhydride copolymers are used in high-heat applications. TIPS is synthesized by copolymerizing styrene and butadiene via the anionic block copolymerization technique. Because most styrenic copolymers are structurally similar to PS, they could be tailored as phase homogenizers in PS/other plastics blends.


**Brominated polystyrene:**


This type of PS is based on HIPS, derivatized with bromine moieties in the polymer chain or modified by incorporating halogenated and analogous additives, rendering it with ignition or flame resistance properties. TV housings and electronic equipment such as computers, printers, and copiers employ brominated PS during fabrication. Some WEEE may contain brominated PS.

## 2. Dealing with PS Wastes

The plastic industry is leaving an indelible mark on society, as it is negatively associated with “plastic pollution”. As the industry grows by producing tons of plastic materials, so are the compounding societal problems in dealing with the enormous waste generated. The future of humanity rests on a more livable environment than is available today, enough to incentivize the search for sustainability and circularity in the use of our planet’s resources. Considering plastics as a resource, the recovery or reuse in a circular economy model proved to be the most challenging for scientists, economic managers, and policymakers. For one, the exponential consumer demand for virgin plastic materials outpaced post-consumer plastic recycling in terms of workable and scalable material recovery technologies and environmental and economic policy adoption. In the last decade, wastes from packaging materials, for example, were treated as a critical concern due to the impact of global warming and other environmental issues, as demonstrated by the growth in research publications concerning this topic. A plethora of mechanical or chemical approaches (Figure 4) for recycling polymeric plastic materials have been proposed [14]. These studies cover a wide range of commonly used plastics, especially polyethylene (PE), PP, and poly(ethylene terephthalate) (PET). However, several approaches could be adaptable for PS wastes as well, and are discussed further in this review. Unlike PS, the recycling of lightweight packaging waste based on polyolefins (PO) separated at source is already state of the art [15]. Furthermore, while several effective recycling strategies exist for PO-type materials, the extensive applications of PS also create a need for a sustainable methodology for dealing with the waste of PS products. These will be covered in the following sections. 

### 2.1. Mechanical Methods

Intuitively, the low environmental footprint of polymeric materials with exceptional performance renders them appropriate resources for a circular economy. However, this is not the case in the current highly linear global plastic industry. Recently there has been a push globally to transition to a circular economy model primarily aiming at the reduction of anthropogenic impact on nature. Realizing a circular economy entails the promotion of the three R’s: reduction, reuse, and recycle, while retaining those plastic materials of the highest value. One such approach is mechanical recycling (MR). The existing MR processes and impediments for recycling PS and other packaging plastics like PET, PE, PP, and PVC have been reviewed through the lens of a circular economy, as shown in Figure 5. The review focused on the global waste management rates in comparison to the European (EU) region, the mechanisms of plastic reprocessing-induced degradation, and the improvement in recycling strategies. Notably, this particular review examined how to improve the blending of the different polymers found in the comingled plastic waste (Figure 6), and finding out potential applications for lower-quality recyclate [16].

Although MR is essentially an environment-friendly and economically sustainable protocol for plastics recycling, the real costs of existing MR processes are still ineffective. Furthermore, MR presents a recovery limit as the mechanical properties degrade with each cycle leading to nth-generation products with inconsistent quality [17]. 

#### 2.1.1. Conventional Mechanical Reprocessing

MR generally involves precleaning, sorting plastics by type, shredding, melt extrusion, and remolding plastics, including PS. Interestingly, HIPS is well suited for MR, since little degradation in mechanical properties is observed after multiple cycles of reprocessing [18]. The basic washing and then drying encompass the precleaning stage. The recycling of plastics is primarily limited by their sorting because their diverse forms, chemical makeup, and formulation are often strongly incompatible, compromising the mechanical properties. MR necessitates manual or automated segregation at the source of the mixed waste stream by the populace or in industrial settings to obtain clean, high-purity mono-material streams to be effective. In turn, this requirement can be the most expensive and time-consuming phase, as most waste streams have diverse compositions or are cross-contaminated with other types of plastic. Composites PS-based materials, for example, are fabricated using more complex components that are oftentimes chemically or physically inseparable. If separation is possible, it mainly involves multiple techniques or processes. Similarly, PS-derived food packaging materials, if separated at all, contain highly contaminated residues. The sorting of WEEE, for example, requires sink and froth flotation and optical and manual separation, among others. In these instances, the recycling of PS is economically cost-inefficient because factors hide the actual value of material recycling.

The viability of the sorting process has been the subject of several studies. Several industrial sorting technologies are available. Some developing technologies still have drawbacks. For example, the carbon black pigment limits the broad adoption of Near-Infrared Hyperspectral Imagery (NIR-HIS) in the sorting process. This pigment is primarily incorporated into thermoplastics like PS as a colorant and UV agent. The presence of pigments such as carbon could impair the proper identification of polymers in a mixed waste stream. Mid-Infrared (MIR-HSI) could be an appropriate alternative methodology for resolving this issue in rare instances. The use of Fourier-Transform Infrared Attenuated Total Reflection (FTIR-ATR) facilitated rapid characterization, lowered misidentification, and enabled specific segregation during industrial sorting [19]. Formulations of the usual WEEE plastics were assessed, with ABS, HIPS, PP, and PE as polymeric matrices, and varying levels of carbon black, titanium oxide, calcite, talc, and flame retardants additives were successfully isolated using this technique (Table 1). The electrostatic separation of mixtures of polymer materials employing a dedicated vision system illustrated in Figure 7 was deemed to be fast, accurate, and effective in isolating the mixtures of PS and poly(methyl methacrylate) (PMMA) [20]. 

Some countries require the sorting of bioplastics and conventional plastics in their waste management programs. However, because the demand for bioplastics is still tiny, few technologies are available for rapid and reliable identification and separation. Such technologies are based on hyperspectral data from a linear spectrometer and a spectroradiometer tuned at ca. 900–1700 nm, the near-infrared region of the light spectrum, shown in Figure 8 [21]. The sensing technology is set on reflectance mode, capable of identifying and sorting out PS, PET, and polylactic acid (PLA) polymer-based materials from each other. The bio-based and biodegradable biopolymer, PLA, could also be sorted.

Sorting plastics as a recycling strategy is cumbersome and can be avoided by introducing plastic blends. Insights into the behavior of waste plastic blends relating to their mechanical properties are essential to avoiding the plastic sorting stage. In Almishal et al., statistical measures (ANOVA) provided some insights into how the mechanical properties of pure PS are affected by an increased ratio of waste PS and PP injected at different temperatures [22]. The study found a dramatic decrease in cost with an increased PS/PP waste ratio at any temperature. A maximum of 30% waste ratio at 200 °C and 220 °C resulted in better mechanical properties.

#### 2.1.2. Landfilling

Mechanical recycling is no longer applicable for complex materials, including composites and multicomponent polymer matrices or contaminated plastic containers such as PS-based food packaging. These materials are very challenging to recycle, and if feasible, they are no longer cost-efficient in economic terms. For technical and economic reasons, unrecycled plastics are intentionally sent to landfills or are incinerated. Waste degradation in landfills typically relies on the anaerobic process. Although the use of a bottom liner and a topsoil cover are standard practices in landfilling, greenhouse gas emissions and hazardous leachate are still of significant concern. Apart from the release of methane, carbon dioxide, other undesirable gases, and aquifer-polluting leachates, landfilling is severely limited by land or space availability.

Remarkably, the rapid growth in the use of EPS compounded by the consumption of package goods outweighs the EPS rate of recycling. The accumulating number of waste EPS entering the landfill is of significant concern, as it takes several decades to break down. This long-term environmental stability of EPS further strains the already limited space for landfills. The gargantuan task of recycling EPS is a misconception, as the most challenging stage boils down to logistics. The waste EPS collection, for example, is uneconomical using most local councils’ standard curbside systems. Moreover, the lightweight nature of EPS masks the actual cost of collection and disposal when bulky EPS wastes are disposed of in skips or bins. Thus, sustainable collection programs are called for in the removal and collection of post-consumer EPS.

### 2.2. Thermochemical Methods

Recycling plastic wastes, especially PS waste streams, is essential to mitigating the disastrous impacts of their current state of disposal in the environment. Like many plastics, PS is recyclable via mechanical reprocessing, thermochemical recycling (TCR), and incineration techniques. MR techniques could be cost-limiting, and they frequently end up with low-value products. However, TCR could be the alternative eco-friendly way to obtain higher-value products from the plastic waste stream. Uttaravalli et al.’s review focused mainly on the engineering and scientific aspects of the many potential applications of waste EPS [23]. This section broadly applies to waste PS in general. 

#### 2.2.1. Conventional Incineration

Incineration involves the combustion of waste, which releases an enormous amount of heat. Before this stage, some recycling methods would have been considered, and the costs for collecting, separating, sorting, and recycling via melt extrusion, among others, make this process expensive. The environmental impact of conventional incineration for municipal sewage wastes is vast compared with other methods like pyrolysis or gasification because of the high volume of carbon dioxide, pollutants like dioxins, and the particulate matter that are emitted. In comparison with MR, conventional plastic waste incineration generates higher CO_2_ emissions but has the potential to recover the associated energy input. The impacts of incineration could be minimized by enhancing the quality of the plastic feedstock through pre-sorting or cleaning and when the byproducts are sufficiently utilized. The installation of incineration equipment that can clean the flue gas effectively, recycle the deposited ash and other residues, and optimize the power and heating cycles to improve energy efficiency are significant additional measures to further the adoption of this technique. 

#### 2.2.2. Conventional Pyrolysis

Pyrolysis is a chemically driven technology that is useful in recovering valuable chemicals from plastic waste. During pyrolysis, plastic waste is broken down in an inert atmosphere at high temperatures [24]. Plastic pyrolysis generates relatively stable gaseous and liquid compounds with high energetic content. Depending on the operational conditions and the nature of the plastic waste stream, the obtained pyrolyzed materials have well-defined distillation points, physicochemical stability, low acidity, and excellent miscibility with traditional fuels. Catalysts used during pyrolysis aid in the facile cracking of large molecules. The catalyst HZSM-5, is well suited to pyrolyzing plastic wastes. The pyrolysis of PS yields a high bio-oil content. The thermal and catalytic processes in the production of pyrolyzed oil were reviewed by several researchers [25,26]. Regarding oil and styrene recovery for PS, a semi-batch reactor outperformed the batch reactor in both thermal (non-catalytic) and catalytic tests at 400 °C [27]. Moreover, a layered catalyst (catalyst separated from PS) was more selective for styrene and yielded higher volumes of oil than a mixed arrangement (catalyst mixed with PS). 

High-value carbon materials could be sourced via pyrolysis when coupled to an in-line catalytic decomposition system after a chemical vapor deposition (CVD) process, leading to the deposition of carbon nanomaterials and H_2_ production. Among the most valuable carbon nanomaterials are graphene quantum dots (GQDs), which exhibit exceptional physiochemical properties in quantum devices. Natural or synthetic carbon-rich compounds are the primary precursors of GQDs. These syntheses are currently conducted at lab scale volume with the procedures requiring hazardous catalysts, acids, or bases.

The pyrolysis of waste plastics is a practical consideration for landfilling, as profits are generated from the accumulated organic matter and pyrolyzed products having acceptable quality. The pyrolysis of plastic fractions of the end-of-life composites (WEEE, vehicles, etc.) and municipal solid waste streams, separated from the organic waste pool, in a batch 5.5 L reactor showed that the presence of PS highly favored forming aromatic compounds, primarily ethylbenzene, toluene, and xylene. With PE, mainly hydrocarbons such as paraffin are produced (Figure 9a). Moreover, with PS, PVC, and PET, the solid content increases, and the liquid yield decreases (Figure 9b) [28].

#### 2.2.3. Microwave-Assisted Pyrolysis

Microwaves broadly correspond to wavelengths of ca. 1 m to 1 mm of the electromagnetic spectrum, assigned to frequencies of 0.3 to 300 GHz [29]. Microwave-assisted synthesis has been proposed as a tool for sustainable chemistry for decades. The advantages of microwave-assisted heating over conventional heating include higher energy efficiency and the capability of non-contact volumetric heating [30]. Microwave-assisted processes facilitated by microwave absorbents or microwave susceptors, for example the pyrolysis of waste streams, can improve the heating rate and reduce processing times. These absorbents are essentially high-loss tangent compounds capable of converting microwave energy to heat. With the appropriate configurations and the proper selection of feedstocks, uniform heat distribution can be achieved in microwave-assisted heating. Microwave-assisted pyrolysis is a nascent technique in dealing with plastic waste to generate bio-oil. 

The microwave-assisted pyrolysis of PS wastes using SiC as susceptors produced the highest yield in bio-oil that can be repurposed for the aviation sector. In one study, ca. 65–85% of oil products consisting mainly of C8–C16 cycloalkanes were produced [31]. Graphene quantum dots (GQDs) were obtained from Styrofoam waste precursors via an acid/base-free, microwave-assisted pyrolysis [32]. The GQDs could be incorporated into the cotton fabric, rendering it self-cleaning and with high hydrophobic properties. The applicability of microwave-assisted pyrolysis as an energy recovery technology has been demonstrated with mixed PS and PP waste streams, decomposing the waste materials entirely within 10 min and yielding bio-oil consisting of gasoline-grade hydrocarbons [33]. The carbon employed as the microwave susceptors were obtained from biomass. Meanwhile, Terapalli et al. used a graphite susceptor in the microwave-assisted catalytic pyrolysis of PS waste in the presence of KOH [34]. Insights on the effects of the catalyst on PS pyrolysis were obtained via design of experiments (DOE) and machine learning (ML) approaches, hinting at an increase in the bio-oil yield consisting primarily of aliphatic hydrocarbons. Some researchers utilized the microwave-metal interaction in pyrolyzing PS through the metal coil inside the pyrolysis reactor [35].

Co-pyrolysis with biomass has already been discussed in detail [36,37]. Waste PS and tea powders were co-pyrolyzed with KOH as a catalyst using microwave heating [38]. Bio-oil and char formation was promoted by the PS and tea powder feedstocks, respectively. Using a catalyst and co-pyrolysis techniques synergically led to generating aromatic and aliphatic hydrocarbons, reducing the presence of oxygenated products. NiFe_2_O_4_ has been employed as the catalyst in a single-step microwave-assisted catalytic breakdown of PS waste, yielding the most solid carbon among other plastics studied [39]. Moreover, the use of NiFe_2_O_4_ generated well-structured carbon nanotubes (CNTs) of relative purity than other catalysts like Al_2_O_3_ and Fe_2_O_3_. On the other hand, using a ZSM-5 catalyst promoted the excessive cracking of algae biomass during the catalytic microwave-assisted co-pyrolysis with EPS or other plastics, leading to higher gas and coke residues [40]. The bio-oil produced predominantly comprised phenolic compounds attributed to algae and monoaromatic hydrocarbon derivatives attributed to EPS.

#### 2.2.4. Conventional Gasification

The gasification of plastic wastes is a thermochemical conversion process that occurs at elevated temperatures in an oxygen-rich atmosphere. Generally, the gasification temperature is much higher than that employed during pyrolysis. In this process, syngas is produced. Syngas is a gas mixture that is used to produce other types of fuel. Syngas can also be directly used as fuel during the incineration of wastes. The exact composition of the product via the gasification process can be controlled by changing the volume of the atmospheric gas or gasifying agent. The final product, however, can be significantly affected by the presence of contaminants. The use of steam or plasma is a developing gasification strategy that has been found to positively affect the sensitivity to contaminants during gasification.

#### 2.2.5. Hydrogenolysis

Hydrogenolysis is a potential approach for the rapid recovery of small molecular hydrocarbons from plastic materials, and enables the processing of waste plastics in complicated conditions. It is a chemical reaction involving the cleavage of a C–C (carbon-carbon) or C–X (carbon-heteroatom) single bond by molecular hydrogen. Hydrogenolysis reactions occur at high temperatures and pressures with limited selectivity, and typically employs a heterogeneous catalyst. The hydrogenolysis of C–C bonds is an essential process in the petroleum industry for waste PO recycling. This section shows the applicability of hydrogenolysis in the recovery of valuable compounds in PS waste, as well as ways to circumvent issues reported in other plastics’ hydrogenolysis. The hydrogenolysis of PS affords the corresponding alkanes and arenes using heterogeneous catalysts. The bimetallic platinum-rhenium silica-supported catalyst, PtRe/SiO_2_, is an ultrawide pore catalyst for PS hydrogenation that has also been employed in isotope exchange reactions [41]. The extent of hydrogenolysis with deuterium exchange was explored at 170 °C under the most severe reaction conditions of 1:1 PtRe/SiO_2_-PS polymer by weight. This study showed that by using this catalyst (Figure 10), the narrow-dispersity-PS was fully converted to poly(cyclohexylethylene) (PCHE) without any detectable hydrogenolysis byproducts.

A homogenous transition metal catalyst that can allow milder reaction conditions in the cleavage of C–C bonds is possible only if the substrate contains strained C–C bonds or directing groups. Employing a borenium complex catalyst (Figure 11a) facilitated the selective hydrogenolysis at the ambient temperature of unstrained C(aryl)–C(alkyl) bonds of alkyl arenes without any directing groups [42]. This catalyst can tolerate a range of functional groups. Waste PS (Figure 11b) was converted into benzene and phenyl alkanes with a mass recovery above 90% using this catalyst during hydrogenolysis, paving the way for the recovery of valuable aromatics from waste PS (Figure 11c).

The hydrogenolysis of PS waste materials at atmospheric pressure and ambient temperature was rapid and efficient using a non-thermal plasma-assisted method [43]. C1–C3 hydrocarbons were obtained at a high yield of >40 wt.%, with ethylene as the main gas product. The ethylene selectivity of this method was >70% within ca. 10 min (Figure 12). The highly active hydrogen plasma significantly affected the reaction kinetics under mild conditions, effectively breaking bonds in the polymer and causing hydrogenolysis.

#### 2.2.6. Catalytic Cracking

Catalytic cracking converts waste plastics into liquid commercial fuel. The catalytic cracking of PS using fly ash as a catalyst yielded a maximum of 88.4% liquid product yield at an optimum temperature of 425 °C for 60 min [44]. The liquid fuel quality of this fly ash cracking of PS is of the gasoline fuel type, with a hydrocarbon range of C3–C24. The experimental data substantially agrees with the Aspen Hysys simulation model (Figure 13) for the PS catalytic cracking in terms of liquid fuel conversion. The model predicted 93.6% of liquid fuel yield for this process in a pyrolytic reactor set at 425 °C.

#### 2.2.7. Dissolution/Precipitation

Some PS-derived plastics like PC/ABS, a copolymer of polycarbonate and acrylonitrile-butadiene-styrene, contain additives like organophosphates, making recycling even more complex. Traditionally, PC/ABS is primarily recovered through MR. However, the effective separation of the plastic and the organophosphate during recycling without degrading the plastic component cannot be accomplished by MR alone [45]. The waste problem is compounded, as some types of organophosphates are hazardous to the environment, especially if they leach out of plastic material. A dissolution/precipitation approach is well suited to effectively separating components in this case. In fact, limonene as a solvent has been used in the manufacture of ABS or HIPS. The naturally occurring liquid limonene is classed as a green solvent and is obtained by extraction from orange peel. Limonene can also undergo free radical polymerization, similar to the styrene monomer [46]. Dissolved recycled EPS in limonene has been shown to increase the mechanical properties of coated kraft paper [47]. Biodiesel, a strong contender for fossil-derived fuels, has been shown as a compatible solvent for PS [48].

Phosphorus-containing PS-based plastic could be recycled efficiently and sustainably by using a switchable hydrophilicity solvent (SHS) such as *N,N*-dimethyl cyclohexylamine (DMCHA) [45]. DMCHA reacts and extracts organophosphates in plastic as the plastic dissolves. From the solution, the PS-based polymer is recovered via precipitation, and the dissolved organophosphates can be quickly recovered because of the SHS property of DMCHA. The recovered materials from this technique were high purity and close to pristine materials per the FTIR results shown in Figure 14.

A solvent mixture consisting of acetone (AC) and ethyl acetate (EA) has been used to dissolve waste EPS [49]. The solvent mixture (Figure 15a,b) was employed as an adhesive binder or asphalt coating film with enhanced mechanical resistance (Figure 15c–e), but was more viscous and dried faster due to the solvent acetone.

A novel closed-loop technology of recycling without generating secondary waste involves the dissolution of waste EPS into styrene monomers [50]. This solution is then subjected to a suspension polymerization technique. No significant differences were observed in the rheological, thermal, and chemical properties of the regenerated polymers compared to a control PS polymer (Figure 16a–d). A process analysis suggests a significant reduction in the impact on the environment and economic viability.

Metallic precursors like Zn(NO_3_)_2_·6H_2_O and Fe(NO_3_)_3_·9H_2_O have been combined with EPS via a solvothermal recycling process in fabricating hybrid nanocomposites with enhanced mechanical properties (schematically shown in Figure 17) [51]. Results from the nanoindentation technique revealed that the hybridized nanocomposites based on Fe_2_O_3_ have the best indentation depth of 0.5 nm under the 20 mN indentation load. The nano hardness was 1.20 GPa with a reduced modulus of 8.20 GPa, an elastic strain recovery of 0.18 GPa, and an anti-wear resistance of 0.025 GPa (Figure 18).

#### 2.2.8. Depolymerization

Depolymerization is another emerging technology for plastic waste upcycling. This technology incentivizes the non-tapping of the remaining fossil fuel reserves, as new monomers used for plastic production can be obtained from existing ones that will otherwise be considered as wastes. However, this requires research investments in catalysts with higher selectivity to break down plastics into monomers. Depolymerization can reduce plastic waste, but its long-term economic viability is yet to be considered, as this pathway’s sizeable and industrial scale is still non-existent. An efficient PS depolymerization process that has the potential for industrial uptake is the use of common table salt with an oxidized copper scrubber, affording a monomer content of >83% [52]. The recovery of styrene facilitates effective re-polymerization of the monomer without affecting the thermal properties as compared to a control PS. Additionally, PS has been shown to depolymerize into a styrene monomer at ambient conditions using mechanochemical processing in the presence of metal-based milling sets made from hardened steel, silicon nitride and tungsten carbide [53]. Depolymerization is attributed to several factors, shown in Figure 19, among which is the macromolecular scission and the action of free radicals generated during milling.

Considerable research has been conducted on the use of clay minerals as catalysts or catalyst supports; however, the similar use of natural clay is still rare. Valášková et al. reported the first use of vermiculites from Brazil and Palabora, which possessed a catalytic effect on PS depolymerization [54]. These two raw clay minerals, VerS and VerP, enabled the thermal recovery of 50.7 and 53.6 mass% styrene monomer (SM), respectively, and 37 and 33.3 mass% pyrolytic volatile oligomers, respectively, consisting of dimers and trimers. Without any vermiculite, the monomer recovery was 54.5 mass%, and the oligomer recovery was 32.1 mass%. The slight differences in yields of the products were due to the presence of the clay minerals. Natural vermiculite characteristics such as differing negative layer charges and the release of water molecules during heat treatment from this interlayer dictate the variable recovery of the monomer, oligomers, and carbon forms. Using Raman spectroscopy, the type of carbon products on vermiculite’s surface could be distinguished in the mixture with amorphous carbon, either coke or glassy carbon.

Several polymers have been shown to be degradable by microwave irradiation [55]. Vast volumes of high-value chemicals like styrene, toluene, and α-methyl styrene have been obtained using this technique at reduced pressure, as well as fuel-grade compounds as byproducts [56] Styrene derivatives and benzene were obtained from mixed plastic wastes undergoing a thermochemical conversion in sub- and supercritical solvents [57]. This process is called solvothermal liquefaction (STL).

#### 2.2.9. Photo-Reforming

Another promising recycling technology that is especially applicable to microplastics present in the aqueous environment is photo-reforming (PR) [58]. PR is driven by sunlight, water, and a photo-catalyst, making it a straightforward and low-energy oxidative process in transforming waste plastics to hydrogen (H_2_) or other high-value chemicals like acetic acid (CH_3_COOH). Among the waste plastics, PS is likely to exist as microplastics in the environment and primarily broken down by UV light, and these PS-based microplastics are the least biodegraded. Because of the chemical makeup of PS, the yields of valuable carbon and H_2_ using photo-reforming were still low.

#### 2.2.10. Sustainable PS-Based Material Design

One of the consequences of society’s rapid population growth, the exponential phase of industrialization, and urbanization is the seemingly unstoppable production of solid waste. Globally, there is now a trend for continuously reusing materials to tackle the waste problem. Scientists, policymakers, and environmental assessors have been tasked with finding a balance between pollution in the environment and the economic advantages of the circularity of sourcing industrial materials. Developing alternative composites based on recycled materials in many industries would prevent the over-exploiting of the world’s natural resources. Many works have illustrated incorporating waste PS streams with other waste materials such as fly ash, other plastics, sludge, glass, and timber dust, aiming for new materials with enhanced physical and mechanical properties. The applicability of this route of recycling methods using various solid wastes in building materials, for example, shows the potential for the energy-efficient production of lightweight materials with increased water absorption and mechanical performance in terms of both compressive strength and plasticity.

Building materials like concrete and bricks have been mixed with other solid wastes as a recycling option. Mortar is among the vital components of cement-based composites, and aggregates significantly affect the mortar properties. Both natural and crushed fine aggregates are commonly used for mortar production in building sectors as energy conservation methods. In addition to aggregates, blocks based on waste plastics and other materials can also be used in construction as a reinforcing strategy [59]. These reinforced plastic paver blocks’ low water absorption properties can then be leveraged for application in the landscaping of footpaths, walkways, plazas, monuments, and waterlogged areas [59].

Tittarelli et al. fabricated lightweight structural mortars with EPS [60]. At the very early stages of curing (2 days), the use of recycled EPS (samples labeled R and HR) led to an improvement in compressive strength development as compared to virgin EPS (samples labeled V). However, the property development was substantially better with virgin EPS than with recycled EPS (Figure 20a) The trend in compressive strength development was found to follow an exponential trend (Figure 20b) It also has to be noted that while the mechanical property development was lower for the recycled EPS compared to a virgin EPS, the increase in tortuosity of the water path in the mortar microstructure (owing to the hydrophobicity and irregular shape of the recycled EPS) leads to a reduced water absorption coefficient for the recycled EPS as compared to the hydrophobic yet regularly shaped virgin EPS (Figure 20c) In addition, there was no significant change in water vapor permeability between the virgin and recycled EPS-based mortars (Figure 20d) Most importantly, a workable mortar with a net economic saving of 25% could be obtained using recycled EPS. Other researchers came to the same conclusions, pointing out that cement mortars with added recycled EPS were lightweight and exhibited higher thermal insulating properties vis- à-vis the sand reference [61]. They attributed the effects to the low specific mass of the samples because of the EPS/cement mixture air spaces along the interface. Rosca developed structural-grade but lightweight concrete with embedded recycled brick aggregate and PS beads as coarse polymeric aggregate [62]. This research found that replacing natural coarse aggregate in EPS/concrete with recycled brick aggregate resulted in 200 kg m^−^^3^ concrete. The mechanical strength is very close to that made with entirely natural aggregate.

A reduction in unit weight and thermal conductivity (critical for thermal isolation application) can be obtained using vermiculite and waste PS as aggregates in mortar. Twenty-five different mortar compositions with varying vermiculite and PS ratios were fabricated into 4 cm × 4 cm × 16 cm dimensions, and their thermomechanical and physical properties were investigated. Using vermiculite and PS in mortar enabled the production of highly porous (up to 67.2%; Figure 21), low unit weight (between 393 and 946 kg m^−^^3^) mortars. The thermal conductivity decreased to 0.09 Wm^−^^1^ K^−^^1^ while the compressive strength varied between 0.57 and 5.89 MPa. It was suggested that mortars with vermiculite and PS could serve as an effective insulating material [63].

As a recycling strategy for the construction industry, plastics derived from WEEE partially replaced the fine and coarse natural aggregates, metakaolin, and condensed silica fume as the reinforcing pozzolans [64]. Replacing with 5, 10, 20, and 30% WEEE in concrete mixtures resulted in a minimum of 25 MPa in terms of structural strength. The improvement in overall performance with WEEE replacement is due to the added pozzolans, increasing the concrete matrix’s density and reducing the interfacial transition zone. Without the added pozzolans, the durability performance and mechanical strength of the WEEE-replaced concrete mixture were inferior to the control mix because of the poor bonding between the natural aggregate and the WEEE particles, owing to the smooth surface of the WEEE particles. This is visualized in Figure 22. Clearly, with 30% WEEP, the concrete’s exposed surface prevented water absorption into the surface of the aggregates, lowering the sorptivity and durability.

PS aggregate has been incorporated into cement, intending to fabricate lightweight concrete masonry units of non-bearing walls having low thermal conductivity [65]. Thermal and mechanical tests with different ages were performed in this work. For concrete mixtures with PS/cement ratio 2.67–6%, the compressive strength (test set up shown in Figure 23) ranged from 4.31–2.67 MPa, the flexural strength ranged from 3.05–1.719 MPa, the density ranged from 1493–1213 kg m^−3^, and the thermal conductivity ranged from ca. 0.91–0.782% compared to the reference mix after aging for 28 days.

EPS waste packaging, together with a cementitious binder, plastic additives, and water, has been demonstrated as an alternative low-cost insulating material [66]. The performance of these insulating materials was comparable to commercially available insulations (thermal conductivity range 0.0603–0.0706 W m^−^^1^ K^−^^1^) and was fire resistant, as shown in Table 2 and Figure 24. This recycling method provided to be an environmentally safe and cost-effective alternative form of home insulation for low-income populations worldwide.

Another recycling strategy of the construction sector is that of developing cement-polymer composites, aiming for the enhanced mechanical durability and the porosity reduction of cement. Simultaneously, any developed product should be stable when exposed to varying quality, origin, and dissolved ions of water. Cement-polymer composites (CPs) with waste EPS exposed in plain, ground, and seawater showed more than 30 MPa in terms of compressive strength after 420 days of immersion (Figure 25). Moreover, the coefficients (K) for corrosion resistance increased significantly in non-immersed samples. Studies by X-ray diffraction (XRD) and SEM in Figure 26 determined alterations in the composite’s microstructure cured for 28 days and after immersion in different types of water for up to 420 days. The enhanced mechanical parameters and the reduced water absorption capacity were claimed to be the result of the depositions of hydration products such as portlandite, calcium silicate hydrate, ettringite, and some salt crystals around the composite pores [67].

The fabrication of sustainable and lightweight self-compacting concrete (LWSCC) with waste EPS has been demonstrated using a water to binder ratio of 0.35 and 500 kg m^−^^3^ binder content, with the coarse aggregate substituted by 0, 40, 50, 60, 70, and 80% waste EPS. An increased waste EPS content led to the increased workability of LWSCCs and decreased strength value, but the compressive strength is within the ACI’s lower limit for structural applications. Compared with the empirical models, the actual strengths met the requirements stipulated in ACI 363 [68]. Hilal et al. observed the same finding on the decrease in compressive strength with the increasing replacement level of EPS [69]. They observed a 3.33 to 50% reduction in values at all ageing durations with their self-consolidating concrete formulations, as shown in Figure 27, compared to a control mix with coarse river aggregate instead of EPS beads.

Recycling wastes produced by the leather industry counters the impact of environmental pollution. Buffing dust from tannery waste was mixed with PS and a blowing agent in 20:77:3 proportions, respectively, to produce thermal insulation panels for the construction sector using a co-twine extruder set at 210 °C. The thermal insulation characteristics of the composite panels are superior to the neat polystyrene boards (Table 3). The composite panels’ thermal conductivity in a density of 300 Kg m^−^^3^ was around 0.029 W m^−^^1^ K^−^^1^ at 27 °C, with a compression strength of 6.25 tons and a water absorption of 7.5%. No degradation was observed in the mechanical properties of the panels, and they were thermally stable from 200 °C to 412 °C. The addition of buffing dust decreased the homogeneity of voids formed in the panels, making them perfect thermal insulation boards in the building sector [70].

The surface protection properties of hydrophobic organosilicon for use in waste PS-based lightweight concrete were analyzed in Barnat-Hunek et al. [71]. The composition of the concrete involved CEM I 42.5 R cement, recycled PS chips of the size 0–2 mm, quartz sand 0–2 mm in diameter, coarse river aggregate of 2–16 mm, and water. The surface treatment of the organosilicon was done using silane and tetramethoxysilane. It was found that the concrete with 20% PS showed increased porosity (25.22%) and thus, an increased absorptivity (14.75%) as compared to the reference concrete (Figure 28). The organosilicon-treated concrete exhibited significantly low surface free energy (SFE). The reduction was as much as 7 to 11 times lower than the SFE of the reference concrete, depending on the type of surface modifying agent used. Contact angle (CA) analysis in combination with frost-resistance (F–T) testing showed that lower SFE translated into lower adhesive properties, higher resistance to water and corrosion from corrosive agents such as salts, and better freeze-thaw resistance. The frost resistance was improved by 54–58%, and absorptivity was reduced by 30% compared to the reference samples. Thus, the use of the treated organosilicon in combination with recycled polystyrene helped fabricate lightweight concrete (LC) with high durability [71]

Functional materials with high value-addition were obtained using recycled PS via a cost-effective and facile dip-coating method [72]. A composite mixture of waste PS and SiO_2_ (PS/SiO_2_) was used to surface coat textiles. The resultant textiles exhibited superhydrophobicity and super oleophilicity. The textiles also showed excellent resistance towards acidic, basic, and saline solutions, high-temperature stimulus, and mechanical abrasion. The PS/SiO_2_-coated textile was also used for the selective separation of oil/water mixtures using either absorption or filtration. The coated textiles also showed a self-cleaning behavior suggested in [72] that could be potentially leveraged towards the effective fabrication of dust-preventing cloths. Demonstrations of each of these, i.e., oil/water separation and self-cleaning fabrics, are shown in Figure 29a and Figure 29b, respectively [72].

The continued over-exploitation of non-renewable resources has made the development of alternatives, such as wood/recycled polymer composites, a highly beneficial research route. Hypothetically, the successful use of waste polystyrene as a binder for the production of wood composites can help avoid the environmental and waste disposal issues bought about by the use of formaldehyde-based adhesives. Such an analysis was conducted in Foti et al. [73], where two types of a panel comprised of 15% and 30% recycled PS were fabricated. The higher PS content (30%) led to an improvement in water absorption (165% and 750%, respectively, for recycled PS content of 15% and 30%) and thickness swelling. However, improvement in the mechanical properties was less pronounced than in the physical properties. The improvement in the rupture modulus was 43.6%, shear strength parallel to the board plane-50%, and glue line shear strength-61.5%. It was proposed that the low viscosity of the recycled PS increased mobility inside the panel matrix, and this manifested in improved penetration up to the adequate depth of the compressed dust particles. An interaction mechanism between the sawdust particles and the PS was developed (Figure 30). Overall, it was concluded that the wooden boards produced using waste PS as a binder could facilitate cleaner and more sustainable production [73].

In [74], a material made of teak sawdust mixed with recycled PS pulp was produced (shown in Figure 31). The properties of this composite material varied significantly with the grain size of the sawdust used. A range of bulk density from 686 to 826 kg m^−^^3^, with a low moisture absorption rate (<15%) and low thickness swell (<5%), was found after soaking for 24 h. Overall porosity was found to vary from 34–43% depending on the grain size of the sawdust. The composite material had a tensile modulus of elasticity ranging from 582 to 1057 MPa, a tensile strength from 2 to 3 MPa, and a Poisson coefficient ranging from 0.14–0.24. The compression modulus of elasticity ranges from 270–470 MPa, and compressive strength from 6 to 9 MPa. This composite material could thus be considered for use as a wood substitute for non-load-bearing products such as shuttering for light construction [74].

#### 2.2.11. Use of Biodegradable Polymers

Bacterial cellulose (BC) is a non-toxic and biocompatible biopolymer. In light-emitting diode (OLED) devices, BC membranes have been used as flexible substrates. However, their semi-transparent appearance tends to limit their performance. That is why BC is copolymerized with various inorganic-organic hybrid materials and fossil-derived monomers to enhance optical transmittance. Nonetheless, the synthesis of these copolymers is not always straightforward. Cebrian et al. demonstrated a scalable and sustainable photonic membrane fabrication method from recycled PS and biopolymer [75]. EPS waste dissolved in transparent *d*-limonene was spray coated on BC membranes to manufacture transparent BC-PS substrates with acceptable performance for OLED applications. These BC-PS membranes exhibited a lower roughness than pristine BC and were found to be an effective substrate in an OLED device, yielding a current efficiency of up to 5 cd A^−1^ (16,000 cd m^−2^) and a power density of ≈2.8 mW cm^−2^.

Based on the guidelines of the Waste for Life initiative, composite materials based on recycled HIPS (obtained from yogurt cups) and paper plastic laminates (from disposable paper cups) were developed [76]. The inherent recycling incompatibility and the sustained use of incineration after first use as a disposal strategy make paper plastic laminates highly unsustainable. Therefore, developing such composites provided a second life to both paper plastic laminates and HIPS. Mechanically speaking, the composite laminates exhibited good performance as indicated by a Young’s Modulus of 1.75 GPa and a Tensile Strength of 21.2 MPa [76].

#### 2.2.12. Biochemical Conversion

Globally, there is a consensus around the addressing of the need for sustainable ways to valorize plastic waste streams. Concepts of reuse and reutilization seen in nature can be used as inspiration From this perspective, nature is the epitome of sustainable waste management. Biochemical conversion could drive the conventional linear-value plastic chain to the sustainability and biodegradable route, creating more value for unutilized post-consumer plastics limited in their MR and TCR suitability. A plethora of similar engineering and technological approaches operating complementarily with natural systems has been presented in a review by Nikolaivitis et al. Among them are sundry MR and green chemical pretreatment processes rendering plastics amenable to biocatalytic and microbial breakdown [77]. Recognized as a more recent undertaking, research on biotechnological-based degradation has the potential to harvest vital monomers and oligomers that can be recirculated to the supply chain.

The European Union launched the MIX-UP project, “Mixed plastics biodegradation and upcycling using microbial communities”, to realize the circular bioplastic economy. In this project, a plastic mixed-waste stream consisting of fossil-based recalcitrant plastics such as PS, PET, PE, PP, and PUR, plus the bioplastics PHA and PLA, served as feedstock for microbial transformations. Selected microbes would enzymatically degrade these mechanically pre-treated plastic wastes, and the mixed cultures would subsequently re-polymerize or regenerate other oligomers, fundamental building blocks, and biomass out of the released monomers. Any residue material unaffected by enzymatic action will be reintroduced into the process after physical or chemical treatment. Protein engineering of known plastic-degrading enzymes would achieve stability, high specific binding capacities, and catalytic efficacy towards a broad spectrum of plastic polymers under high salt and temperature conditions. Another project’s focus is the discovery or isolation of new enzymes that could break down these recalcitrant polymers. MIX-UP will formulate and produce enzymes tailored to specific plastics [78].

Microorganisms, insects, and their associated gut microbiota can mediate PS biodegradation, and this approach has been presented as a potential alternative to MR and TCR methods. PS waste is constantly accumulating in the environment, especially Styrofoam, the most common PS-based litter. Thus, efficient and environment-friendly ways of recycling are needed. Biodegradation has been demonstrated as a possible means of recovery for waste PS and other plastics.

Yellow mealworms are the larvae of the beetle *Tenebrio molitor*. When fed with a diet of oatmeal and polymers such as PS, PE re-granulates, and lignocellulose, profound changes in the biochemistry and digestive tract microbiota of mealworm larvae were observed. Mealworm larvae can chemically modify PS, and they developed best on PE re-granulates with minor mass loss during the transition from one developmental stage to another. A diet based on lignocellulose was the least favorable for larval development. The protein content in mealworm larvae benefitted the most from a PS diet, but the fatty acid content in the insects fed these wastes was lower in terms of quality as a food source than in the control insects. Specific microbial cultures, protozoa, and various biochemical activities were induced by each diet, suggesting different strategies of survival and varying mechanisms of assimilating feedstocks [79]. The bacteria present in the larvae of *T. molitor* after feeding with PS foam as the only carbon source in their diet were identified and isolated by Machona et al. The biodegradation of PS was also confirmed from the frass egested by mealworms containing minuscule PS residues. The three gram-negative bacteria obtained from the guts during isolation, Klebsiella oxytoca ATCC 13182, Klebsiella oxytoca NBRC 102593, and Klebsiella oxytoca JCM 1665, were believed to be responsible for PS biodegradation [80]. However, Matyja et al. discovered that *T. molitor* larvae fed with PS had difficulty completing their life cycle, raising doubts about using mealworms effectively for PS recycling [81]. They inferred from the Dynamic Energy Budget (DEB) model that the type of PS feedstock affected the organism’s growth trajectory and metabolism in 91-day monitoring. A decrease in reserve density and the larvae’s reaction to the insufficient food supply primarily drove the changes in the larvae’s development when fed with PS.

Superworms from the species of Tenebrionidae, *Zophobas atratus* larvae, had a high rate of survival when fed with 43.3 ± 1.5 mg PS or 52.9 ± 3.1 mg LDPE/100 larvae per day over 28-day monitoring. However, their body fat content decreased. The larvae can biodegrade PS more effectively than LDPE. The ingested PS and LDPE were turned into microplastics (particles size of 6.3 and 5.9 μm, respectively) but not nano plastics. *Citrobacter* sp. was detected in the gut when PS and LDPE were fed. Microbial functional enzymes, including aryl esterase and serine-hydrolase, were associated with PS or PE degradation [82]. In the study of Arunrattiyakorn et al., three bacteria, *Pseudomonas* sp. EDB1, *Bacillus* sp. EDA4, and *Brevibacterium* sp. EDX, were present in the gut of *Zophobas atratus* larvae [83]. A 30-day PS incubation with each strain led to biofilm formation on the PS surface films. SEM, FTIR, and WCA characterizations confirmed that all isolates could degrade PS, with the *Brevibacterium* sp. EDX (GenBank MZ32399) being the most efficient PS-degrading strain.

Both larvae of superworms *Zophobas atratus (Fab.)* and yellow mealworms *Tenebrio molitor (Linn.)* can survive on sole plastic diets (PS or PUR) in 35-day monitoring, showing that dominant microbiomes drive the microorganism’s propensity for intake of specific plastics [84]. The superworms survived 100% on all diets with decreased weights over 20 days, while the yellow mealworms survived 84.67% on the PS diet and 62.67% on the PUR diet, with increased weights on both diets. The cumulative plastic consumption of superworms was 18-fold that of mealworms on a PS diet and 11-fold that when on a PUR diet. In terms of mg/g-larvae, a higher PS consumption rate was exhibited by the superworms, but both have similar levels on a diet of PUR. A PS diet in superworms increased the relative abundances of unclassified *Enterobacteriaceae*, *Klebsiella*, *Enterococcus*, *Dysgonomonas*, and *Sphingobacterium*, while *Hafnia* was upregulated in yellow mealworms. In a diet of PUR, *Enterococcus* and *Mangrovibacter* were dominant in superworm guts, while unclassified Enterobacteriaceae and Hafnia were the dominant gut microbiome in yellow mealworms.

Greater Wax Moth larvae (*Galleria mellonella* L., Lepidoptera, Pyralidae), mainly inhabiting the northern hemisphere, can also biodegrade PS and single-layered PE. Barrionuevo et al. were the first to observe wild *G. mellonella* larvae in Argentina consuming biaxially oriented PP and PE-based silo bags [85]. In their study, the insect’s larvae with a mean size of 25–30 mm were fed with EPS, single-layered LDPE, triple-layered PE (SB, for silo bags), BOPP, and beeswax as a control diet. EPS and PE (both LDPE and SB) were consumed mainly by the larvae moreso than BOPP, yet they still emerged as adults. The pupal stage was faster for larvae that consumed plastics than for control, regardless of the amount and type of plastic. The multi-kingdom synergy between bacteria and a fungus is active in the gut of *Galleria mellonella* L. In a follow-up study by Barrionuevo et al., Argentina’s greater wax moth larvae were put on a 7-day diet of PS, PE, and beeswax as control. Sequencing revealed an increase in two *Pseudomonas* strains in the larvae gut microbiome. No differences in the composition of the fungal communities were observed between diets, except for their relative abundance [86].

Microplastics in the environment are now a grave concern, significantly affecting the aquatic environment because of their long biodegradation time. This problem is further exacerbated by environmental erosion and temperature, further reducing the size of the microplastic particles. Eventually, sea waves will deposit them on the beaches, which poses more threats to humans. *Tenebrio molitor* and *Galleria mellonella* larvae were used by Pinchi et al. to biodegrade microplastics collected from the Azul beach in Ventanilla [87]. The collected microplastics in this work were determined to be EPS, PET, and PVC. The microplastics were fed to larvae of *Tenebrio molitor* and *Galleria mellonella* for periods of 5, 10, and 15 days. After 15 days, a higher level of biodegradation for all plastics, i.e., 54.2% with 30 larvae of *Tenebrio molitor* for the EPS and 34.4% with 30 *Galleria mellonella* larvae for PVC. Direct in vivo evidence for the associated metabolic pathways and the influence of gut microbiota in the *Galleria mellonella*’s larvae was further investigated via the enforced injection of 0.5 mg PS microbeads per one larva of *G. mellonella* (Tianjin, China) and general-purpose 2.5 mg PS powders per larva [88]. The PS microbeads have a molecular weight of *Mn* 540 and *Mw* 550, while the PS powders have an *Mn* of 95,600 and an *Mw* of 217,000. The PS microplastics were broken down and digested independently of gut microbiota in *G. mellonella.* The styrene oxide–phenylacetaldehyde and 4-methyl phenol–4-hydroxybenzaldehyde–4-hydroxybenzoate metabolic pathways were suggested as the processes involved during biodegradation. Enzymes produced by *G. mellonella* drove the biodegradation of PS efficiently. These enzymes have yet to be identified and isolated.

Similarly, insect larvae have been demonstrated as an eco-friendly biodegradation treatment for WEEE plastics. In the study by Zhu et al., nine WEEE and virgin plastics were chosen as feedstock for *Galleria mellonella* and *Tenebrio molitor* larva [89]. The larvae of *G. mellonella* favored consuming virgin plastics over WEEE plastics consisting of PS or rigid PUR. This is possibly because of the increased levels of chlorine or metals in the WEEE plastics. SEM and FTIR analyses of larval frass confirmed the partial biodegradation of WEEE plastics. Waste HIPS in powder and non-lumpy form improved the uptake by *G. mellonella* larvae. In turn, *G. mellonella* larvae have a decreasing preference for pristine plastics under individual-plastic-fed mode: Rigid PUR > phenol–formaldehyde resin > PE > PP > PS ≈ PVC. PE consumption by *G. mellonella* larvae is higher than that of PS, while *T. molitor* larvae showed the opposite trend.

### 2.3. Assessments of PS Recycling Methods

Since most plastics do not naturally degrade, increasing plastic recycling rates is critical to mitigating plastic accumulation in the environment. An uptake in recycling could lower the dependence on fossil resources, ultimately reducing the global plastic waste stream. Plastics such as PS are currently derived from carbon sources mainly associated with the extraction of fossil fuels. Fossil-derived processes are energy intensive and recognized as sources of greenhouse gas (GHG) emissions that exacerbate global warming conditions. Therefore, the increasing worldwide consumption of plastics considerably impacts climate change. This foreseeable impact is reason enough to develop recycling technologies, where the continued reuse of available plastics is highlighted, limiting additional plastics from the world’s scarce sources of raw materials. Importantly, a comparative assessment of these technologies within defined system boundaries is necessary, which could include life cycle (LCA), techno-economic, environmental impact, and future regulatory compliance.

In most European countries, the ultimate destination of plastic waste is usually either a landfill or an incinerator. The collection and recycling of mixed lightweight packaging (LWP) waste in Germany consisting of PS, PE, PP, and PVC underwent a techno-economic and environmental assessment against the country’s state-of-the-art mechanical recycling as the baseline [90]. The assessment involved a combination of mass flow analysis (MFA) and LCA for multiple scenarios in the MR and the use of rotary kiln pyrolysis technology, a TCR method, and a combination thereof considering the processes’ global warming potential (GWP), reports as [CO_2_e], cumulative energy demand (CED) in MJ/kg, carbon efficiency [%], and product costs. In turn, a hybrid of MR and TCR by pyrolysis approach for LWP was deemed to be in compliance with both the German and European Union’s target rates of recycling in 2018. An estimated 0.8 and 2 million metric tons/year was allocated for the circular economy that could instead go to the incinerators with the adoption of this combined technology. In comparison with the German baseline, the calculated GWP was 0.48 kg CO_2_e/kg input, the CED was 13.32 MJ/kg input, and the cost was 0.14 €/kg input, with a carbon efficiency of 16% higher. In Belgium, Civancik-Uslu et al. [4] assessed by LCA the impact of the MR and TCR of sorted household waste consisting of PS, PE films, PP, and mixed PO rigids against incineration with embodied energy recovery as the baseline. The assessment revealed that MR or TCR recycling approaches were more beneficial than the baseline for all fractions of investigated plastics, with the highest benefit achieved in recycling rigid PS. TCR approaches avoid the new synthesis of virgin styrene monomer, while MR prevents the new production of virgin PS. MR generally has a less net impact on the environment than TCR if the recycled products are suitable 1:1 replacements for virgin materials. This caveat holds because the circularity loop for MR is shorter than TCR. The TCR and MR of PS yielded a GWP of −1580 and −3096 kg CO_2_e/ton input vs. the baseline of 2494 kg CO_2_e/ton input.

The economic feasibility of implementing MR for plastic waste such as PS, PP, PE films, and mixed POs was examined in [1]. As stated earlier, the economic incentives for recycling plastic packaging depend predominantly on product price and yield. This incentive is jeopardized at low oil prices, substantially decreasing the prices of virgin plastic products. Even with a discount rate of 15% for 15 years, MR is not profitable without any changes in governmental policy [1]. In several other localities, LCA also pointed out the cost-benefits of recycling waste EPS [91].

A novel proposal for managing waste WEEE is illustrated in Figure 32, encompassing sorting, dissolution/precipitation, extrusion, catalytic pyrolysis, and plastic upgrading. The environmental impact of this proposal was determined by LCA and vetted with the currently adopted schemes in Europe that included conventional MR, TCR, improper dumping, open burning, and exportation of WEEE to developing countries. The proposed WEEE plastics management scheme is environmentally sustainable, increasing the annual volume of wastes from 390 kt/y up to 530 kt/y sent to recycling, decreasing residues from 360 kt/y up to 60 kt/y sent for incineration, and reducing the potential impact of all the midpoint categories under analysis, i.e., up to 580% for that of Global Warming. Aside from the Global Warming impact as a sustainability criterion, the proposed schemes scored well in the Carcinogens and Non-Renewable Energy categories, with improvements of 60% and 17%, respectively, vis-a-vis the current ideal scenario in Figure 33. If a dissolution/precipitation process that could recover polymers like HIPS, ABS, and PC was adopted, improvements of 246% for Global Warming, 69% for Non-Carcinogens, and 35% for Carcinogens were projected, even without considering upgrading and catalytic pyrolysis. The new scheme became less advantageous if waste exportation outside of Europe was included in the analyses, suggesting that for the management of WEEE to be sustainable, limiting the improper treatment of exported wastes should be of utmost priority [92]. Another LCA study of recycled HIPS and ABS from WEEE suggests that these recycled polymers can return to the intended function even with a decline in some properties as long as compatibility issues during blending with other polymers are addressed [93].

Sixty-four percent of all plastic manufacturers in Ecuador are in Guayaquil City. Hidalgo-Crespo et al. [94] used LCA to determine the effects circumvented by the circular economy scenario for the city in terms of waste EPS materials. This assessment found that recycling EPS plastic waste significantly benefited the environment [94]. EPS-based food containers were assessed in a cradle-to-grave approach. The entire LCA was focused on different valorization paths: a circular economy closed-loop (container-to-container) proposal with an electricity share of 2019, and another with a future electricity share (2027), and a linear economy (container-to-landfill). Among these three scenarios, a circular economy model for recycling post-consumer EPS waste with proposed 2027 electricity share (scenario C in Figure 34) had lesser environmental impacts in terms of terrestrial/ marine eutrophication, acidification, ozone depletion, and land use, scoring −21%, −24%, −28%, and −31%, respectively.

## 3. Toughening Approaches of Recycled PS as an Upcycling Strategy

Upcycling can be achieved by improving the mechanical and other associated characteristics of a recycled PS. This may be done by blending with other polymers or using additives and other techniques, and are enlisted in this section. For the thermomechanical upcycling of waste PS and other plastics, several chemical additives and approaches that effectively enabled adhesion between the interfaces of differing components leading to phase homogeneity have already been reviewed and studied [95,96,97].

### 3.1. Use of Compatibilizers

Recycling is a time- and cost-intensive undertaking. These reasons are enough to discourage the already few existing recycling industries. Apart from determining the appropriate recycling methods, the chemical compatibility between the diverse nature of waste streams needs to be considered. Otherwise, phase separation of the different components could lead to low-quality products. To circumvent phase separation, chemical modifiers are introduced during the re-extrusion of plastic wastes. The modifiers are also called coupling agents, compatibilizers, or performance enhancers. Compatibilization is a viable option to negate the exacerbating costs of upcycling PS waste. Regarding the chemical structure, some compatibilizers are oligomer/polymers synthesized as block (*b*), graft (*g*), and random (r) copolymers. Each compatibilizer’s chemistry is tailored to the mixture’s thermodynamic interactions between immiscible components. The compatibility improvement in blended polymers involves incorporating another substance (the compatibilizer) which can facilitate the cost-effective conversion of plastics to high-value products by optimizing their miscibility and stability [98,99]; hence, compatibilization is an upcycling technique.

It has been reported that PS fraction in WEEE plastic is the predominant polymeric component, necessitating recovery of this polymer in WEEE [100]. Recycling of this fraction without separating it into components was demonstrated by Grigorescu et al. using a melt compounding technique [101]. In [101], a blend of styrene-butadiene-*b*-copolymer (SBS) and hydrogenated and maleic anhydride-modified SBS (SEBS-*g*-MAH) acted as the compatibilizers or impact modifiers. It was claimed that the PS-derived composites from this process have improved UV and flame resistance (while no such testing was conducted), and the physical and mechanical properties were comparable to HIPS. The general trends in mechanical properties obtained are shown in Figure 35.

It is possible to fabricate ABS/HIPS blends recovered from WEEE. Hirayama et al. further demonstrated that the recovered ABS/HIPS blend could have controllable mechanical properties using compatibilizers such as styrene–butadiene–styrene (SBS) and styrene–ethylene–butylene–styrene/styrene–ethylene–butylene (SEBS/SEB) [3]. SBS and SEBS/SEB as compatibilizers had significantly affected the morphology of HIPS/ABS. Adding the compatibilizers changed the size and shape of the dispersed PS-derived phases (Table 4), resulting in mechanical property changes in the polymer. Incorporating virgin polymer in the blends also affected the morphology.

In Jaidev et al.’s blending work, recycled PVC and ABS and recycled HIPS were recovered from uninterrupted power supply (UPS) housing and television housing, respectively [102]. The phase homogeneity in the PVC/ABS and PVC/HIPS blends was inferred from morphological, thermal, and mechanical analyses. At similar composition of r-PVC, the blend with r-ABS, has better mechanical properties than the blend with r-HIPS, as shown in Figure 36. Blending r-PVC with r-ABS at a 70:30 ratio yielded an impact strength of 250 J m^−1^, 200% higher than the r-ABS (r-PVC = 0%). Furthermore, the study posited a mechanism of cross-linking in Figure 37 between ABS and PVC, leaching chlorine atoms.

### 3.2. Addition of Tyre Crumbs or Rubber

Recycling tires at end-of-life can facilitate the recovery of functional raw materials for future use that is sustainable and environment-friendly. Valuable materials like rubber, carbon black, steel, and fibers can be recovered from discarded tires. Devulcanized rubber tires (DVR) as fillers could be a sustainable way of recycling waste rubber tires and PS. DVRs have been used as fillers for thermoplastic thermal insulators [103]. They have been incorporated in powder form into PS from 0–50 wt.% using a melt extruder. Superior properties have been observed with composites consisting of less than 40 wt.% DVR filler, with thermal conductivity in the range of 0.0502 to 0.07084 W m^−1^ K^−1^, at densities ranging from 462.8 to 482.32 kg m^−3^, are shown in Figure 38. In addition, the compressive strength was in the range of 11.66 to 7.47 MPa, and the flexural strength was ca. 40.4 to 19.26 MPa. Hashin & Shtrikman’s conduction models (series and parallel) validated the thermal conductivity results. Furthermore, the DVR filler treatment with a base had a positive compatibilizing effect, significantly improving the mechanical and thermal stability of the composites.

Ground tire rubber (GTR) with particles lower than 200 microns has been blended with several polymers. In fact, six potential low-requirement insulator applications, provided in Table 5, were determined that meet the rigorous requirements of the Spanish Association for Standardization (UNE) and the International Electrotechnical Commission (IEC) [104]. Although PS/GTR blends were part of this study and found to be sufficiently insulating, with minimal decrease in their mechanical properties (tensile strength and elongation at break), the overall properties exhibited by PS/GTR blends do not conform with the strict requirements set by UNE and IEC for electrical applications, paramount of which was the elongation at break.

### 3.3. Ternary Blends

The blending of polymers has the distinct advantage of interfusing the properties of each component polymer. Binary blends of polymers have been commonly employed, and only recently did ternary blends catch the attention of material scientists. Ternary blending leads to a variety of micro and nano morphologies. The structuration at this domain level has been demonstrated to improve the mechanical properties of the employed matrix thermosets [105] and thermoplastics [106,107]. An exploration of the ternary blending of PS has been carried out by several researchers [108,109,110]. However, only a few reports exist where PS was upcycled from the waste stream. The straightforward ternary blending approach would encourage PS-based recycling.

Jaidev et al. identified the polymers from keyboard waste aided by the keyboard parts’ embossed resin identification code (RIC). They formulated a ternary blend from the recovered polymers [100], intending to improve the thermal, mechanical, and morphological characteristics. The recovered polymers PS, ABS, and HIPS from the WEEE have impact strengths of 29 J∙m^−1^, 42 J∙m^−1^_,_ and 20 J∙m^−1^, respectively (Figure 39). When a ternary blend contained PS as the major phase, the impact strength increased to 66 J∙m^−1^. An improvement in the ternary blends’ thermal stability from 380 °C to 396 °C was observed, with a negligible effect on the glass transition temperature (*Tg*). The blends’ salami morphology enabled a brittle to ductile transition, improving the observed characteristics.

Relying on the RIC and Fourier transform infrared (FTIR) spectroscopy to sort waste keyboards components, in a similar study formulating ternary blends of varying levels of PS, ABS, and HIPS, the optimum tensile strength, flexural strength, and impact strength of 35 ± 3 MPa, 65 ± 3 MPa, 45 ± 3 J∙m^−1^ were determined, respectively, for the ternary blend [111]. Morphological and thermal analyses of impact fractured specimens revealed that the blends were homogeneous. A 98% degradation of the blends was also observed at 700 °C using thermogravimetry analysis (TGA). No significant changes in the properties of blends were noted, paving the way for the elimination of multiple sorting of plastic components, thereby reducing the cost aspect with this upcycling approach.

### 3.4. Ionic-Crosslinking

The structure of PS consists of bulky and rigid phenyl groups dangling on the polymer backbone, and the segmental interaction along the chain is weak and unpolarized. These characteristics of PS result in its poor impact toughness and fair heat resistance. Cross-linking with ionic moieties has been demonstrated to circumvent these effects. Cai et al. used zinc dimethacrylate (ZDMA) to demonstrate ionic cross-linking into recycled ABS/HIPS blends [112]. ZDMA could self-polymerize, forming poly-ZDMA (PZDMA) particles. Cross-linking could occur by graft polymerization with butadiene on the recycled HIPS or ABS chains of ZDMA. The ionic interaction between unsaturated carboxylate and Zn^2+^ in Figure 40 improved the compatibility between recycled HIPS and ABS. An improvement in the blends’ storage and loss modulus, impact strength, and tensile strength were observed. The notched impact strength at four wt.% ZDMA was 133% higher than that of blends without ZDMA. The polybutadiene (PB) phase was well dispersed into the recycled HIPS/ABS blends, as seen by using SEM. The excellent compatibility of recycled HIPS/ABS blends in this work was significantly affected by ionic cross-linking.

Another use of Zn-based ions for cross-linking PS was demonstrated by Yu et al., although neat PS was employed. It is expected that the methodology could also apply to the PS waste stream. Yu et al. aimed to enhance the heat resistance during the melt blending of PS by incorporating a PS ionomer derivative, the high-ion-content Zn-salt poly(styrene-*ran*-cinnamic-acid) (SCA-Zn) [113]. This ionomer derivative needed to be of dense ionic cross-linking but of low molecular weight to enhance the blend’s *Tg* and ease of processing during melt blending. Consequently, SCA was synthesized by copolymerizing styrene with 20 wt.% cinnamic acid. SCA was then melt-reacted and neutralized by excess ZnO to ca 21,000 g mol^−1^, deemed the half-critical weight average molecular weight (Mw) for this purpose. Varying levels of SCA-Zn were incorporated with PS during melt blending with good compatibility. The phase-separated morphology of the 60/40 PS/SCA-Zn blend resulted in higher heat resistance.

### 3.5. Addition of Polymers, Organic and Inorganic Fillers

Adding other oligomers or polymers, organic and inorganic, is an effective toughening method for the polymer matrix. Hayeemasae et al. prepared a composite based on recycled PS and eggshell powder (ESP) [114]. As the secondary filler, calcium carbonate (CaCO_3_) was added to reinforce the composite’s performance. During the extrusion of the ESP/CaCO_3_ hybrid PS composites, the torque decreased with the increasing percentage of CaCO_3_. Furthermore, increasing the percentage of CaCO_3_ in the hybrid composites enhanced Young’s modulus, impact strength, tensile strength, and elongation at break. SEM images in Figure 41 revealed greater homogeneity in the dispersion of CaCO_3_ into the recycled PS matrix than that of the ESP. Moreover, CaCO_3_ provided better heat resistance than ESP. This study suggested a composite formulation of 10/10 wt.% CaCO_3_ and ESP with recycled PS exhibiting a balanced set of thermal and mechanical properties. Graphite oxide (GrO), having many agglomerated flakes, was added as a filler (0.1 wt.%) by Ferreira et al. into the brittle PS matrix [115]. The agglomerates toughened the PS matrix instead of acting as defects. The amorphous polymer then exhibits mechanical properties close to the semicrystalline polymer, causing higher deformations and lower modulus. The low filler level enabled the organized agglomeration in a stacked way, rendering them with reinforcement and super lubricity effects. Ab Ghani et al. used graphene nanoplatelets as a filler in PS and found that 15 wt.% of styrene butadiene rubber (SBR) as the toughener exhibited the best mechanical properties [116]. Physical reinforcement of the brittle PS has been demonstrated using polyhedral oligomeric silsesquioxane (POSS) [117]. POSS is chemically inert, and only physical interactions are possible with PS; however, different structural types of POSS (lamellar and dumbbell-shape) significantly influenced the overall properties of fabricated composites. Chemical interactions with PS are facilitated by incorporating appropriately functionalized POSS, as in the case of POSS cages with vinyl or vinylidene moieties. The mechanical properties of SiO_2_-PS composites with grafted chain conformation were dependent on the P/N ratio [118], where P is the degree of polymerization of the matrix and N is the degree of polymerization of the grafted brush. High composite strength is expected at N ~ P, with swollen grafted chains. The grafted chains at N < P are expected to be collapsed, facilitating particle pull-out. Thus, tuning the P/N ratio could increase strength or strain energy density.

## 4. Conclusions and Future Outlook

Sustainable and economical potential routes of recycling for post-consumer polystyrene (PS) waste streams remain a challenge different from that of other plastics, i.e., PET and HDPE, because of value propositions. This work is a review of the problems arising from PS waste and focussed on the overall loop that starts from the production cycle, proceeds to waste generation the eventual reuse of PS waste streams. The impact of the PS waste, its recycling methodologies, and analyses of the PS cycles of life were presented. Profitable value chains for recycling mixed PS-based plastic have hardly progressed to more commercial stages today. Cognizant of the realization that the PS supply chain is virtually impossible to transform self-sufficient overnight completely, several methodologies and approaches have been detailed in this review in pursuit of a circular economy. One thing remains certain: hybrid approaches of mechanical and thermochemical PS recycling techniques would be required to accomplish this task. It is recognized that mechanical recycling at the source is necessary for any feasible approach, with the thermochemical methods giving an upcycling gain. Recycling at source is counterproductive with PS wastes, particularly EPS and WEEE, without any legal and regulatory enforcement to the public and private enterprise. An aftermarket for upcycled PS will only thrive when governments incentivize consumers/producers to adopt a more circular business supply chain specific to PS. It is incumbent upon governments to make legal arrangements mandating businesses to provide take-back services, especially for EPS and WEEE, because the waste collection of these materials is rate-determining for the PS supply circularity models. In turn, increased legal pressure should encourage the broad adoption of promising technologies elucidated in this to the commercial phase.

Several mechanical approaches detailed in this review, using PS wastes as drop-in additives in other applications, provided meager upcycling gains in view of the originally intended applications. MR mainly produces materials vastly different from the original to be functional as upcycled products. Hence, MR alone will not be a practical approach to recycling. Although demonstrated as a feasible recycling strategy, EPS in construction materials still lacks commercial scale, and no prototype exists that validates the results. Furthermore, most studies have not considered the long-term environmental consequences of the approaches when buildings/construction are demolished in the future. Regarding future technologies, research on building construction with recycled PS should also focus on how these materials are amenable to 3D printing, given that PS has excellent radiation resistance. When feasibility is confirmed, habitat construction in space could be a new direction to further the upcycling research, especially with regard to how the new constructions withstand radiation in space.

From a usage perspective, TCR approaches yielded more significant benefits in recycling, as the upcycled products can be used as initially intended. TCR, especially in combination with MR, could be the most viable way to recycle PS. However, in most recent studies relating to TCR, the toxicity effect of styrene as the recovered monomer, especially in the form of a volatile organic compound (VOC), was excluded. Some TCR approaches are currently sources of blue hydrogen. Therefore, these potential approaches, i.e., pyrolysis, should also focus on carbon capture technologies to complete an all-sustaining system of sourcing H_2_. When the global warming potential associated with these technologies is further lowered, then it is only a question of time before the availability of green hydrogen from an infinitely recycled carbon-based precursor could complement those sourced from renewable energy. Biodegradation using mixed microbes is potentially another viable strategy for recycling plastics. This technology is amenable to analogous vertical farming, already being trialed on a pilot scale. However, research should also focus on containment and long-term exposure protocols to lessen the environmental effects. The biodegradation using insects should be reconsidered, as most works do not consider PS ingestion’s long-term effects and toxicity. Although the ingestion of microplastics by these living organisms to degrade plastic has been explored, the volume of work for the ingestion of nano plastics is lower. The adverse ecological effects of nano plastic are well-detailed in the literature and should not be discounted in the biochemical conversion approach mentioned in this review.

Considering recycled PS applicability to frontier applications, life cycle analyses and techno-economic assessments could be made comprehensive by adding these parameters. In most cases, waste transport to faraway locations is not considered, which can significantly affect the approach’s viability after considering the benefits of hydrogen fuels in future transportation. The carcinogenetic effects of the styrene monomer on humans and ecology were highlighted in the literature; perhaps this is the time to stop the use of this monomer and virgin PS plastics and simply recycle all that is available. From this perspective, the approaches of recycling PS waste to recover the monomer are further incentivized. However, to mitigate the deleterious effects of styrene, a closed-loop system is necessary; this is where the many gaps in the literature concerning styrene recovery approaches exist. One of the methods that could be explored to circumvent this dilemma is to use flow chemistry. This would involve a system where waste PS is injected as feedstocks in a series of flow chemistry reactors to be deconstructed to the monomer or oligomers, capturing them and the associated byproducts for an in situ re-polymerization or other material recovery treatment. The re-polymerization could be directed toward manufacturing broad-usage and high-value styrenic oligomers that can be further employed in the MR and TCR strategies mentioned. Researchers in pursuit of sustainability should give this proposal or similar adaptable methodologies that provide multi-faceted benefits their immediate attention.

## Figures and Tables

**Figure 1 polymers-14-05010-f001:**
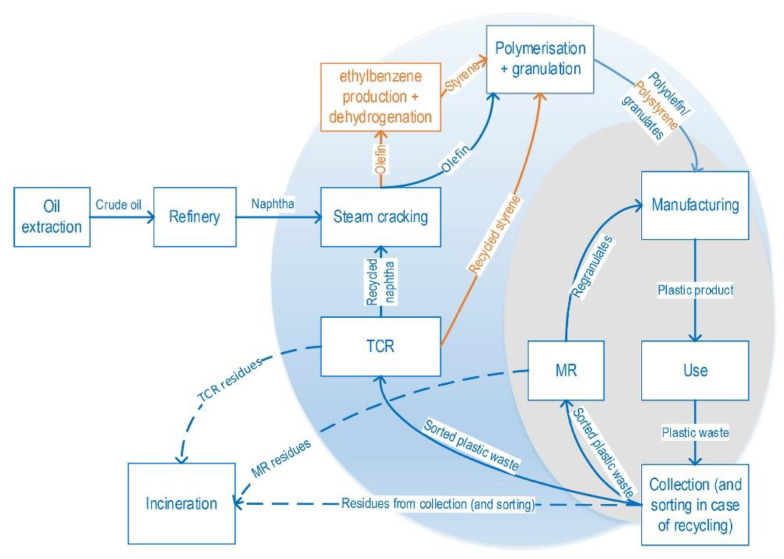
The linear and circular economy based on PS. Reprinted with permission from ref. [4].

**Figure 2 polymers-14-05010-f002:**
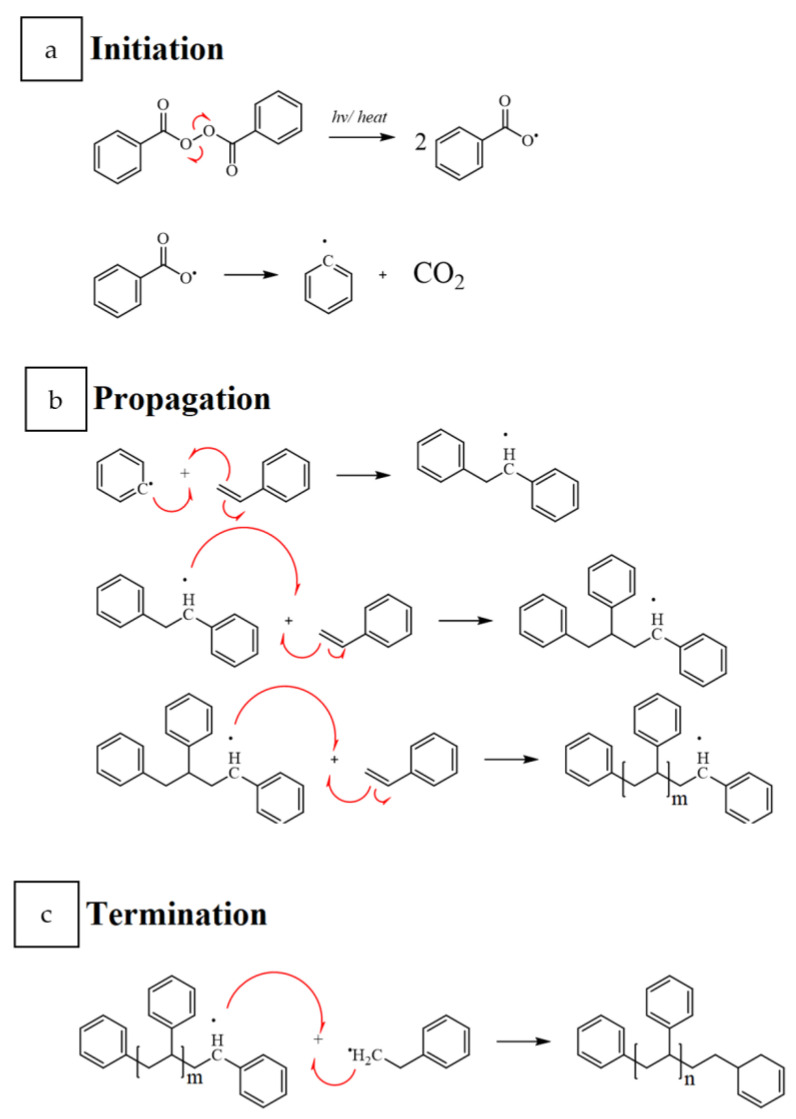
Benzoyl peroxide-initiated polymerization of PS.

**Figure 3 polymers-14-05010-f003:**
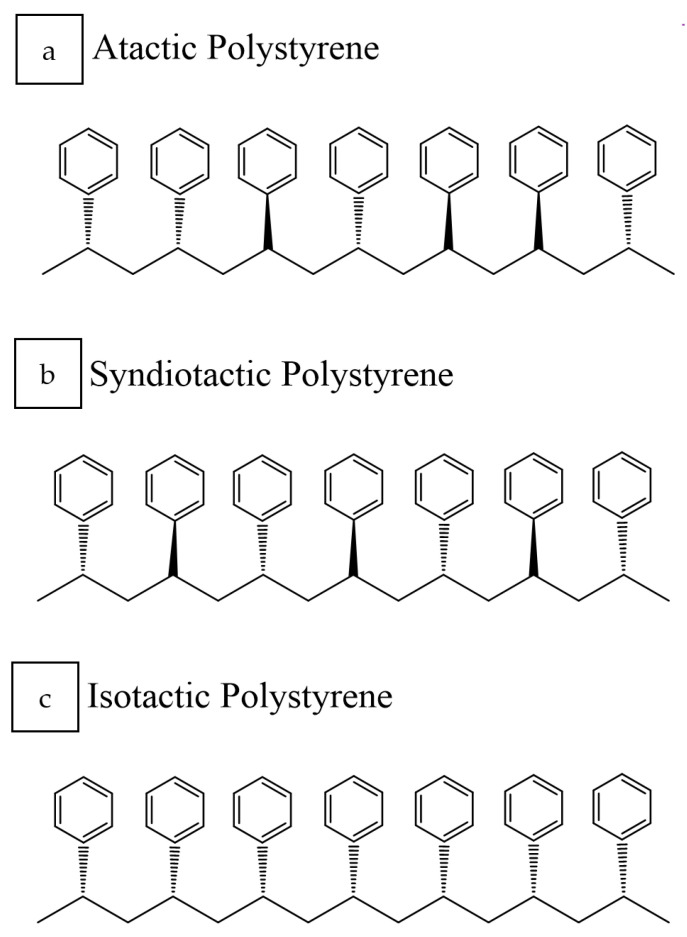
Different forms of PS.

**Figure 4 polymers-14-05010-f004:**
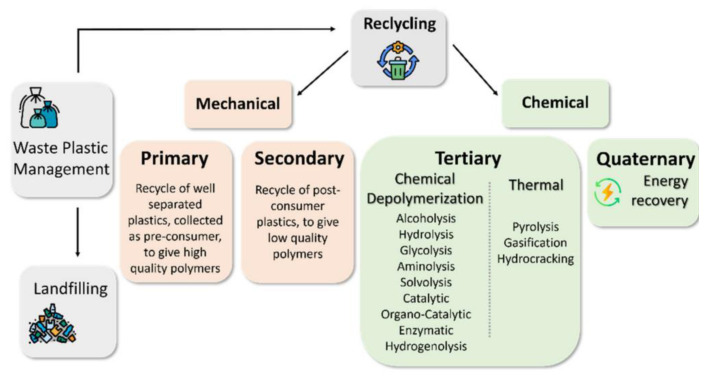
Overview of plastic recycling methodologies. Reprinted with permission from ref. [14].

**Figure 5 polymers-14-05010-f005:**
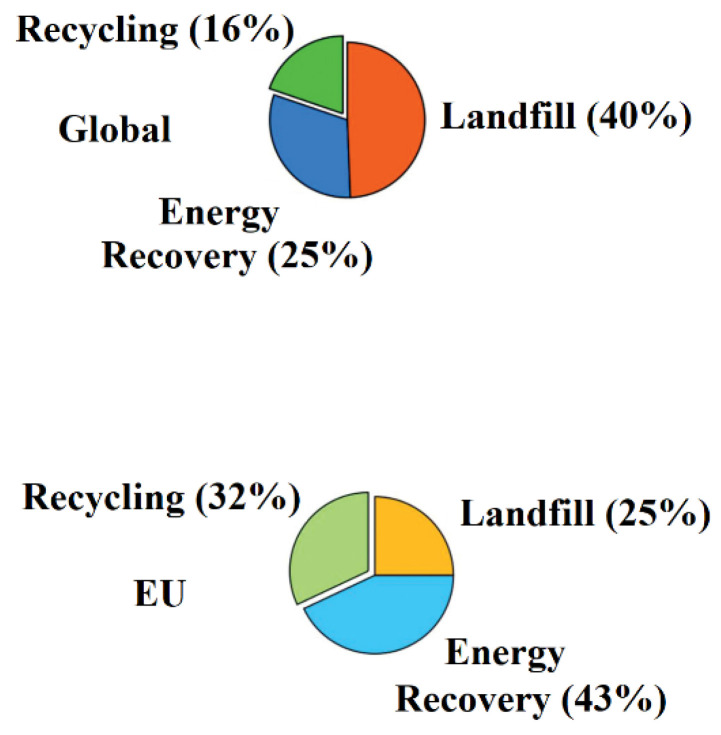
Global and EU waste management rates. Reprinted with permission from ref. [16].

**Figure 6 polymers-14-05010-f006:**
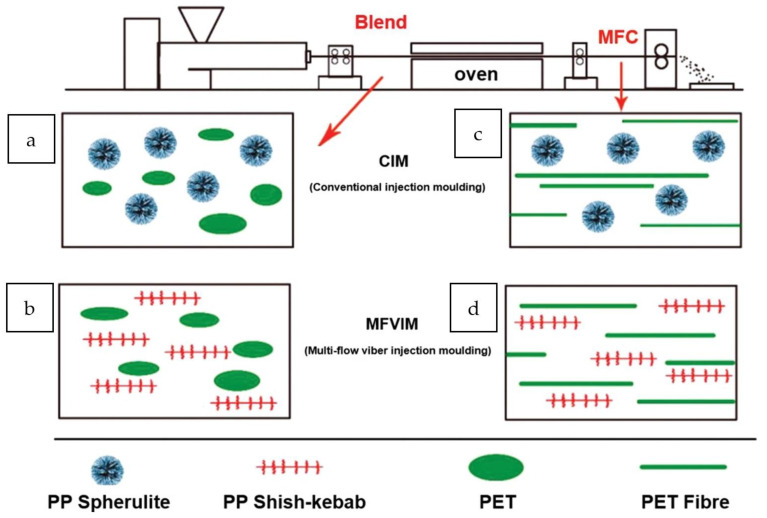
Using novel blending approaches to upscale properties of recycled PO blends using microfibrillar PET (**a**), (**c**) Using conventional injection molding (CIM), (**b**,**d**) Using multi-flow viber injection molding (MFVIM). Reprinted with permission from ref. [16].

**Figure 7 polymers-14-05010-f007:**
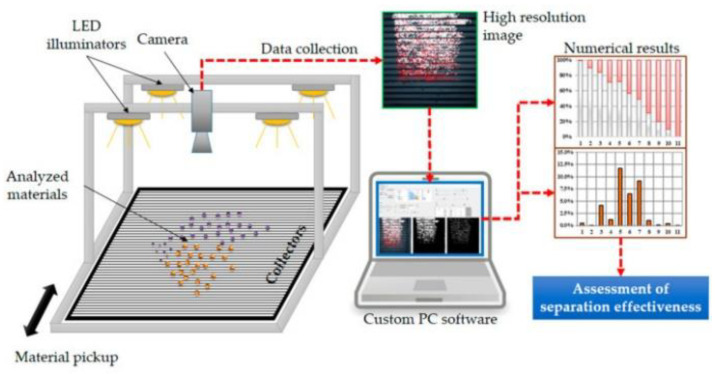
Illustration of the vision system for assessing the effectiveness of the electrostatic separation process. Reprinted with permission from ref. [20].

**Figure 8 polymers-14-05010-f008:**
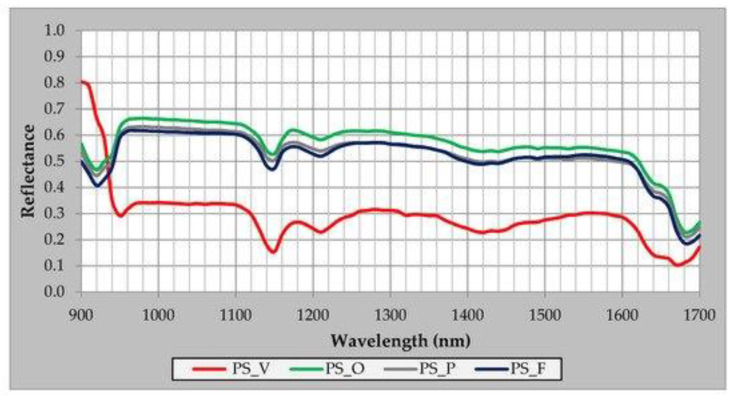
The NIR spectra of PS samples were obtained by a linear spectrometer, where (V) are virgin particles, (O) represents the original shape of waste, (P) are large pieces, and (F) are flakes. Reprinted with permission from ref. [21].

**Figure 9 polymers-14-05010-f009:**
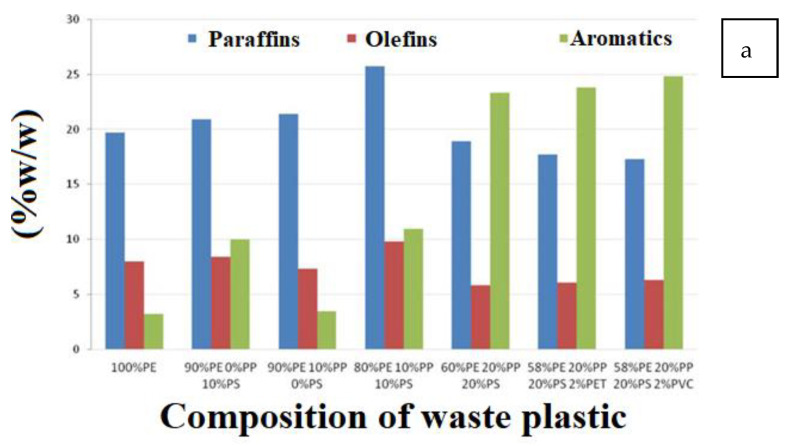
(**a**) Chemical composition (in weight %) and (**b**) Solid, liquid and gas (in weight %) yield of product obtained from pyrolysis of WEEE and MSW streams. Reprinted with permission from ref. [28].

**Figure 10 polymers-14-05010-f010:**
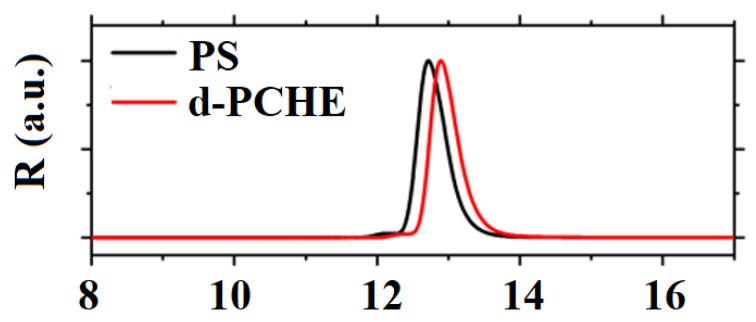
Size exclusion chromatograms before (black curves) and after (red curves) hydrogenolysis with deuterium exchange (PtRe/SiO_2_ catalyst to PS at 1:1 (w/w) loading, 17 h, 170 °C). PS (black curve) was fully saturated, forming PCHE (red curve) before and after the reaction. Reprinted with permission from ref. [41].

**Figure 11 polymers-14-05010-f011:**
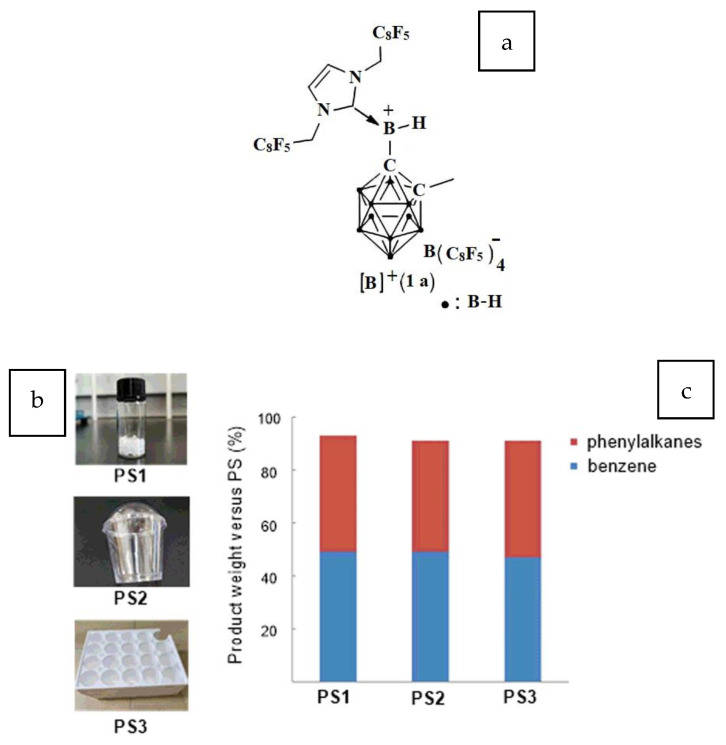
Hydrogenolysis of PS with (**a**) catalyst, (**b**) PS1: laboratory-grade polystyrene; PS2: polystyrene single-use cup; PS3: EPS foam (0.1 g PS + 0.05 mmol catalyst, 10 mL o-C_6_H_4_F_2_, 5 bar H_2_). (**c**) Phenylalkane and benzene yields. Reprinted with permission from ref. [42].

**Figure 12 polymers-14-05010-f012:**
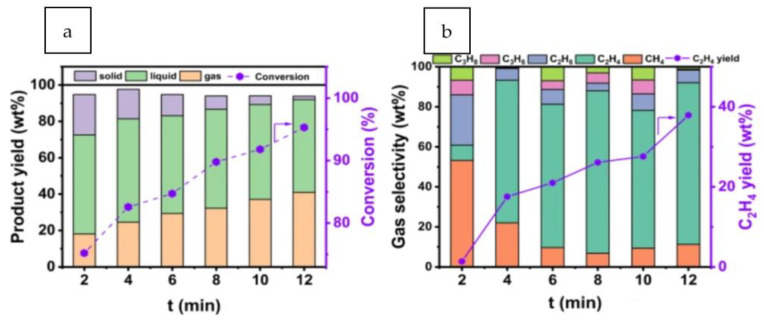
(**a**) Product yield and (**b**) gas selectivity of hydrogenolysis of post-consumer PS. Reprinted with permission from ref. [43].

**Figure 13 polymers-14-05010-f013:**
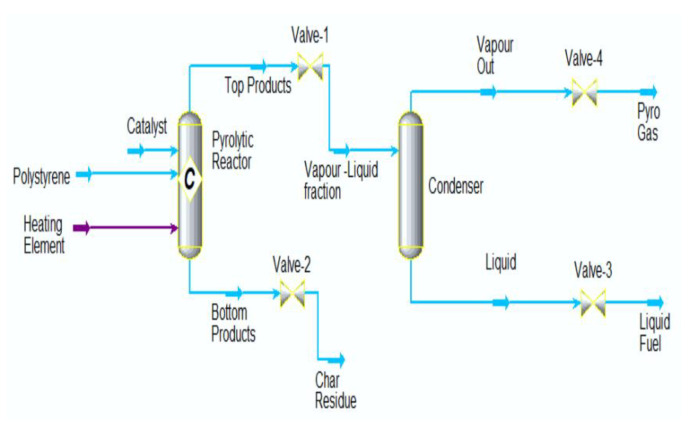
Simulation environment of the Aspen Hysys simulation model for the catalytic cracking of PS. Reprinted with permission from ref. [44].

**Figure 14 polymers-14-05010-f014:**
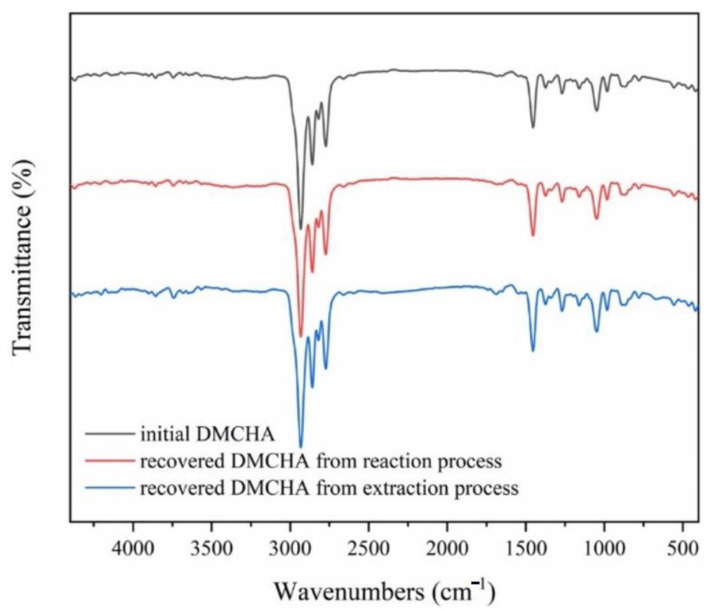
FTIR spectra of initial DMCHA and recovered DMCHA. Reprinted with permission from ref. [45].

**Figure 15 polymers-14-05010-f015:**
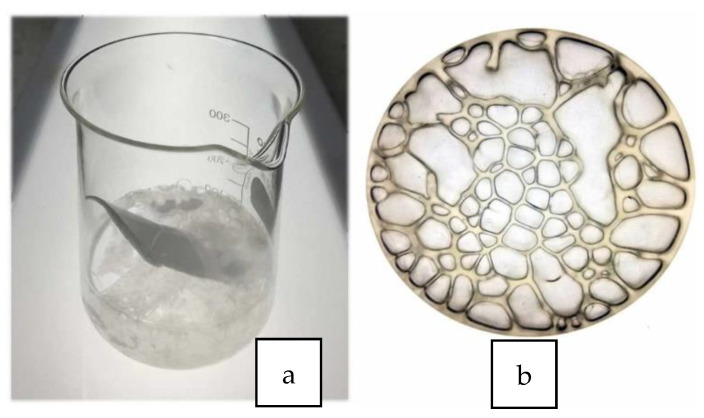
Drying and dissolution of waste EPS. (**a**) dissolution of EPS in acetone/ethyl acetate mixtures; (**b**) bubble formation after drying the waste, (**c**) binder load strength, (**d**) apparent strength shear; and (**e**) stickiness. Reprinted with permission from ref. [49].

**Figure 16 polymers-14-05010-f016:**
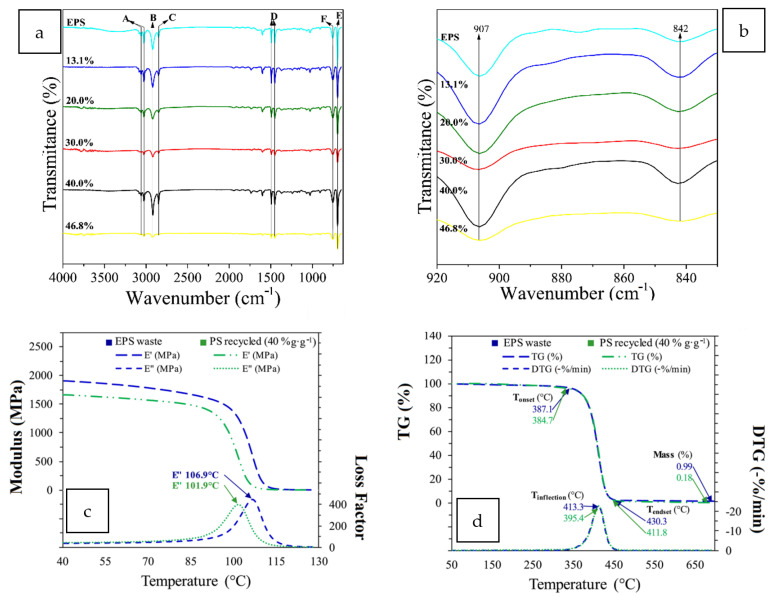
Comparison of (**a**,**b**): Chemical (via FTIR), (**c**): Thermal (through % weight loss thermogravimetry TG and first derivative differential thermogravimetry DTG data), and (**d**): Mechanical (Storage Modulus E’ (in MPa) and Loss Modulus E’’ (in MPa)) characteristics of regenerated PS with source EPS. Reprinted with permission from Ref. [50].

**Figure 17 polymers-14-05010-f017:**
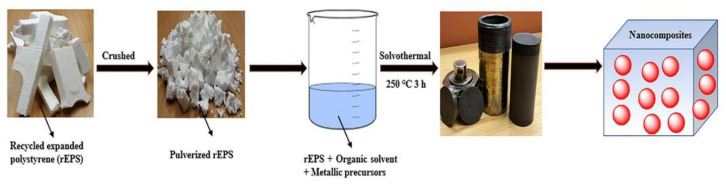
Synthesis scheme via the solvothermal method for hybrid nanocomposites. Reprinted with permission from ref. [51].

**Figure 18 polymers-14-05010-f018:**
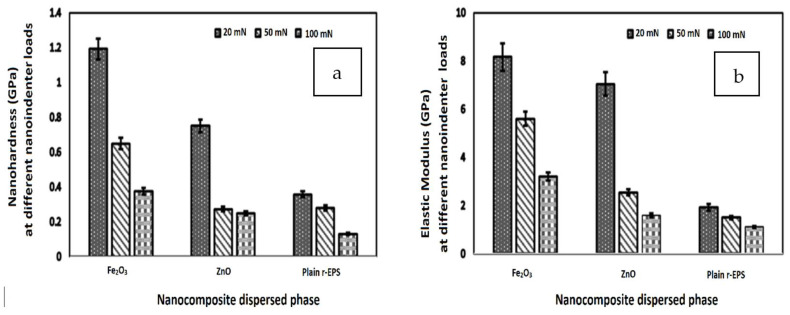
(**a**) Nanohardness and (**b**) elastic modulus under indentation loads of 20, 50, and 100 mN for Fe_2_O_3_ nanocomposites, ZnO nanocomposites, and recycled EPS composite. Reprinted with permission from ref. [51].

**Figure 19 polymers-14-05010-f019:**
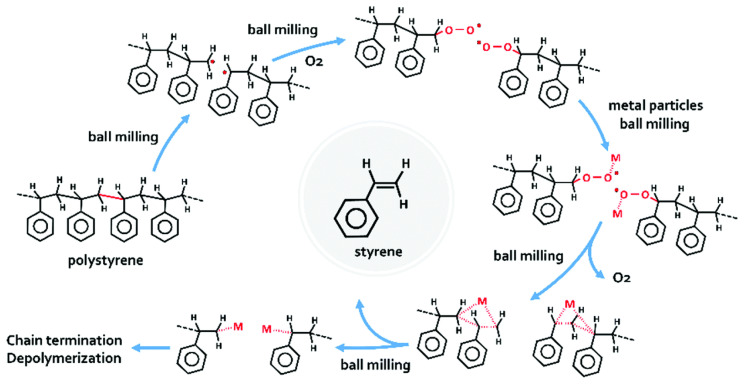
Proposed mechanism of polystyrene depolymerization in salt and oxidized copper scrubber. Reprinted with permission from ref. [53].

**Figure 20 polymers-14-05010-f020:**
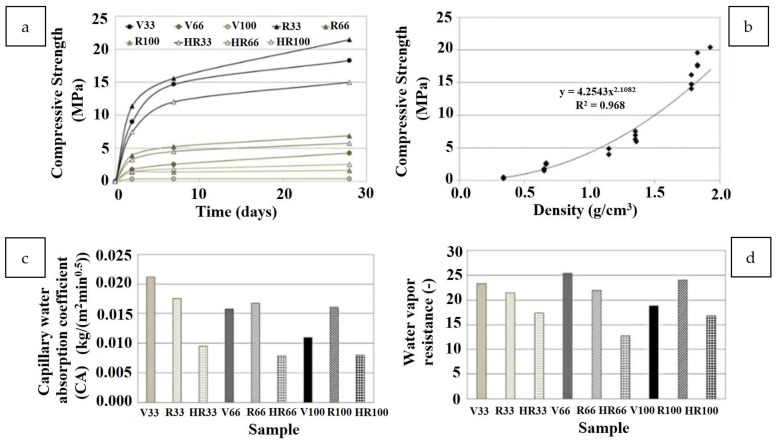
Compressive strength development with (**a**) time and (**b**) mortar density of EPS-based mortars, (**c**) Capillary water absorption coefficient (CA), and (**d**) water vapor resistance (μ) of lightweight EPS-based mortars. Samples V implies that virgin EPS replaces the sand volume, R implies that recycled EPS replaces the sand volume, and HR implies that recycled EPS with 0.5% silane agent replaces the sand volume, while the numbers indicate the extent of volume replacement. Reprinted with permission from ref. [60].

**Figure 21 polymers-14-05010-f021:**
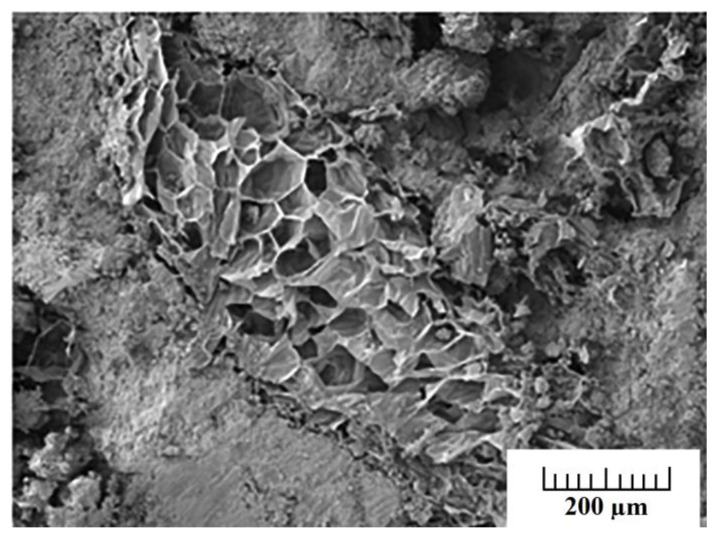
Scanning electron microscopy (SEM) images of mortar produced with EPS aggregate. Reprinted with permission from ref. [63].

**Figure 22 polymers-14-05010-f022:**
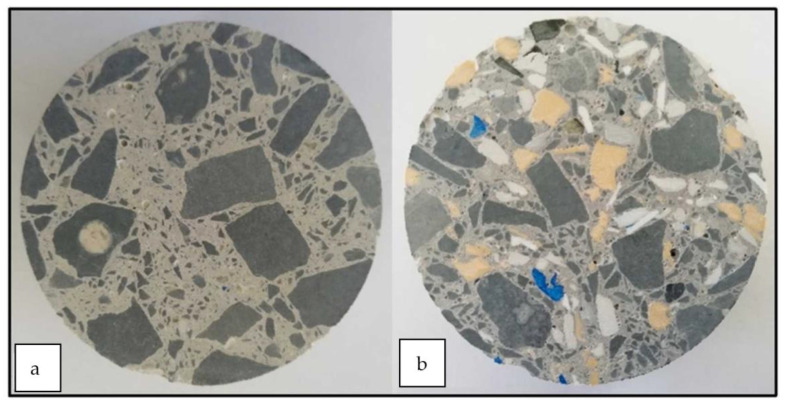
Exposed surfaces of the concrete disc (**a**) without WEEE (Control mix) and (**b**) with 30% WEEE (PC-Blend-30). Reprinted with permission from ref. [64].

**Figure 23 polymers-14-05010-f023:**
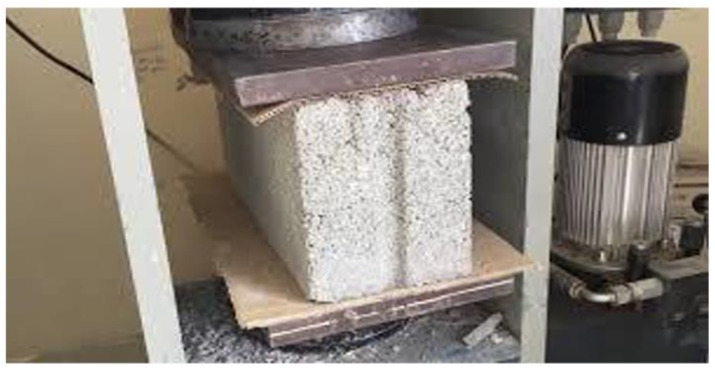
Compression testing of PS/cement composite material. Reprinted with permission from ref. [65].

**Figure 24 polymers-14-05010-f024:**
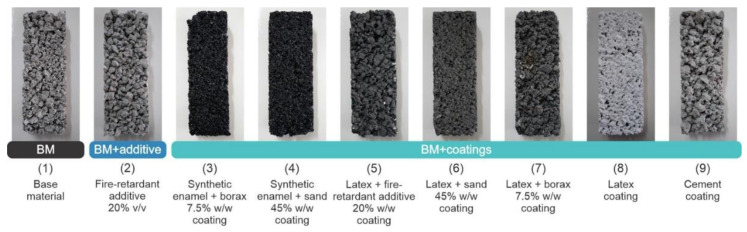
Types of samples used in the single flame source test. Reprinted with permission from ref. [66].

**Figure 25 polymers-14-05010-f025:**
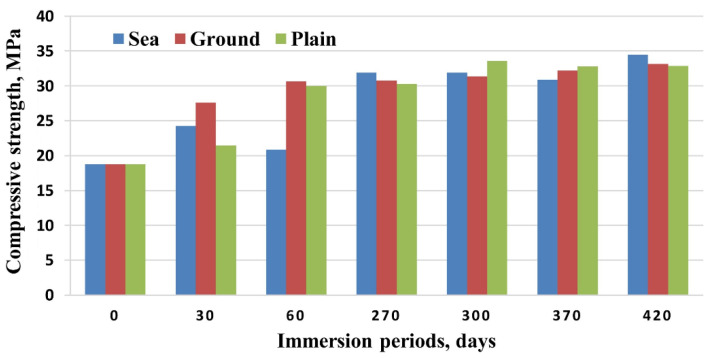
Impact of immersion periods on the compressive strength of the cement-polymer composites in plain, ground, and seawater. Reprinted with permission from ref. [67].

**Figure 26 polymers-14-05010-f026:**
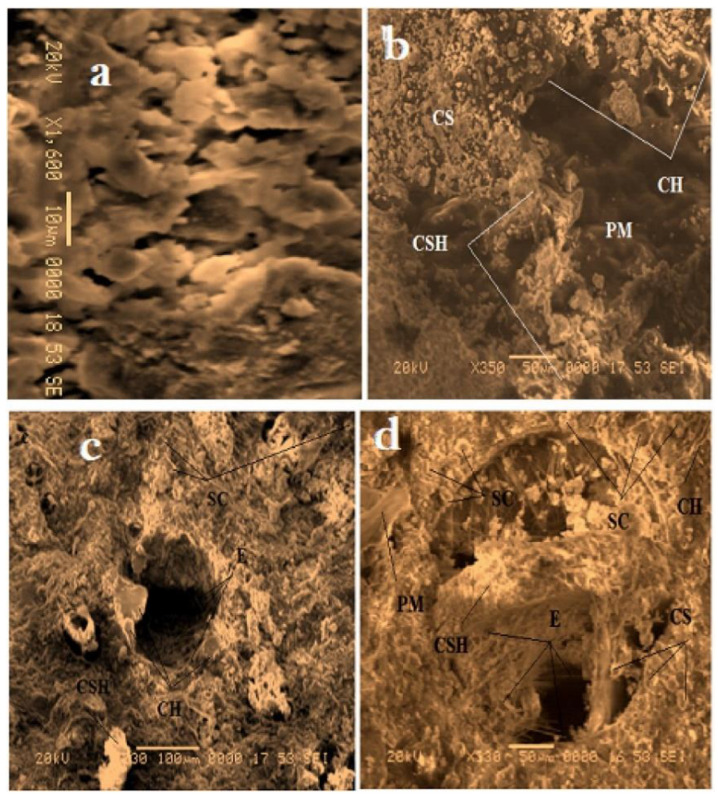
Microstructure of the composite material under various conditions (**a**) before immersion and after immersion in (**b**) plain water, (**c**) groundwater, and (**d**) seawater for 420 days. Portlandite [CH], calcium silicate hydrate [CSH], salt crystals [SC], ettringite [E], and polymer matrix [PM]. Reprinted with permission from ref. [67].

**Figure 27 polymers-14-05010-f027:**
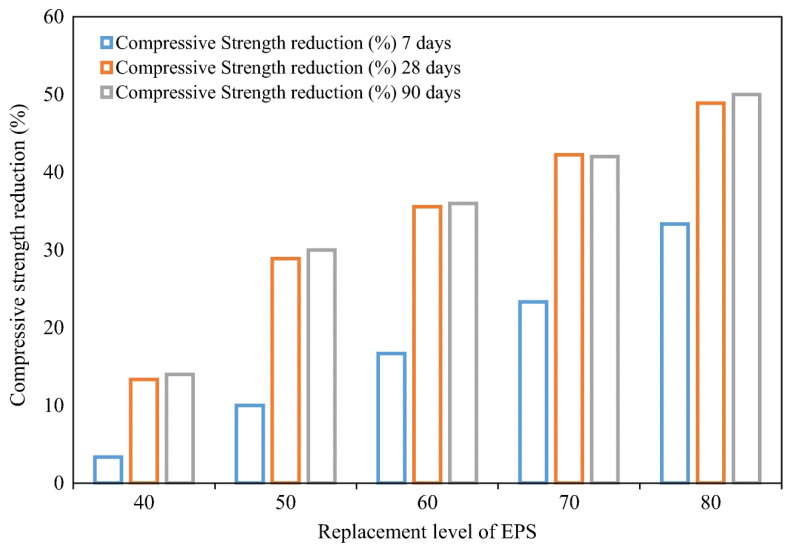
Variation in the reduction of compressive strength (%) in self-consolidating lightweight concretes as the typical coarse river aggregate was replaced with 40, 50, 60, 70 and 80% EPS beads. Reprinted from ref. [69].

**Figure 28 polymers-14-05010-f028:**
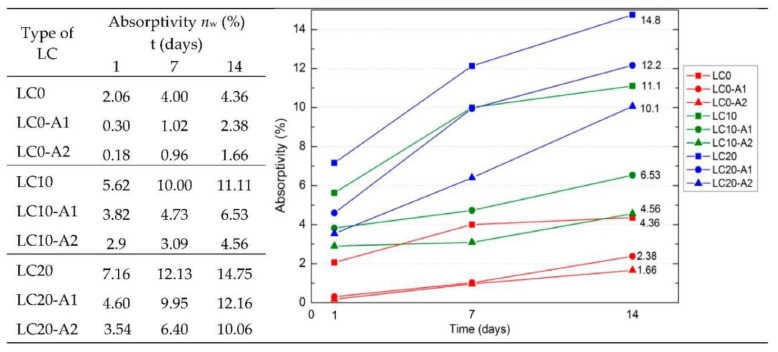
Water absorptivity of recycled PS-based concrete. Reprinted with permission from ref. [71].

**Figure 29 polymers-14-05010-f029:**
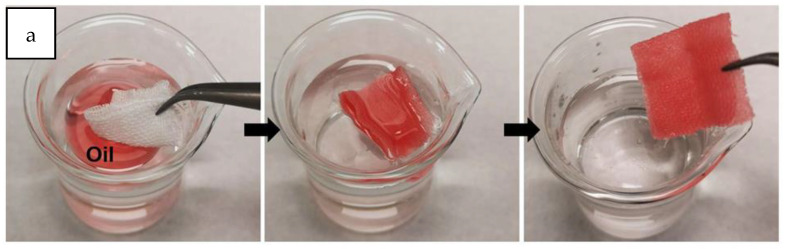
(**a**): Oil absorption process by PS/SiO_2_-coated textile from the water surface, (**b**): Surface self-cleaning effect of PS/SiO_2_-coated textile. Reprinted with permission from ref. [72].

**Figure 30 polymers-14-05010-f030:**
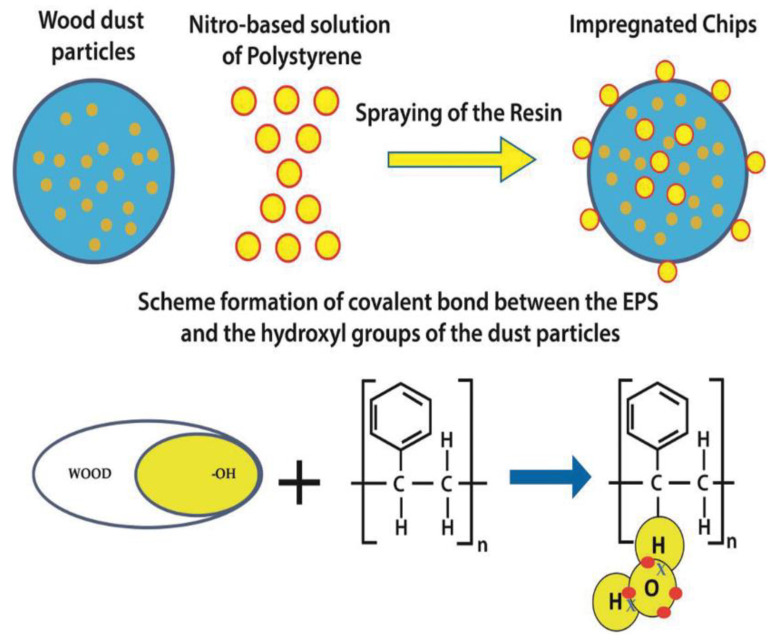
Mechanism of covalent bond formation between the recycled EPS and the sawdust. Reprinted with permission from ref. [73].

**Figure 31 polymers-14-05010-f031:**
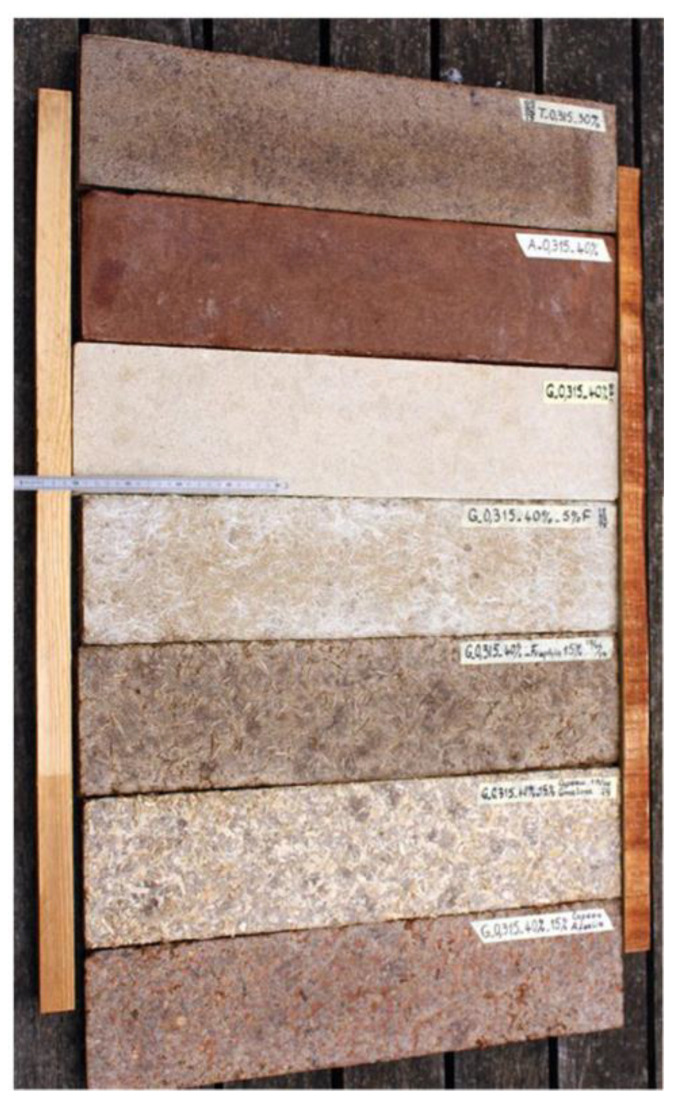
Sawdust-based panels of different wood species. Reprinted with permission from ref. [74].

**Figure 32 polymers-14-05010-f032:**
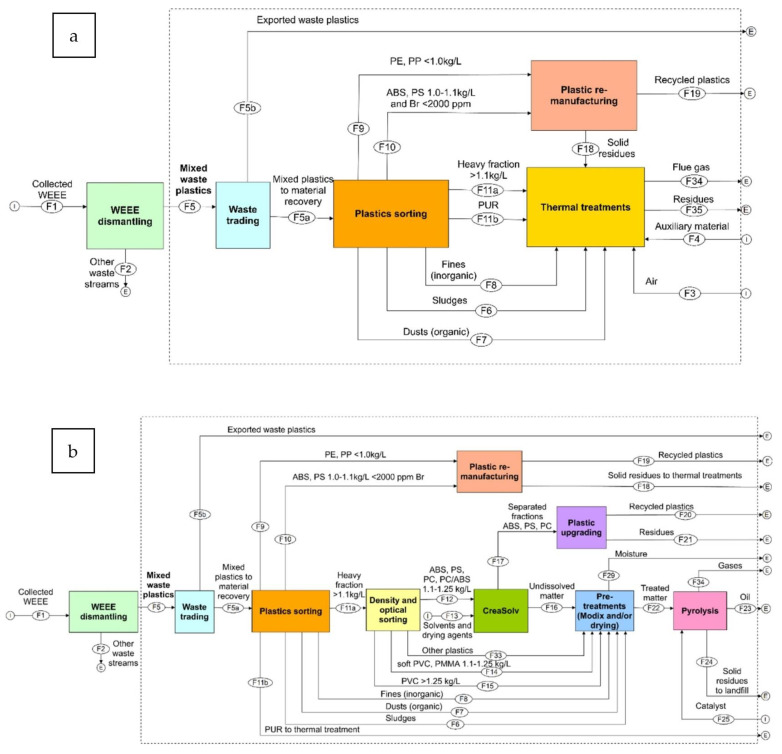
Comparison of the (**a**): current and (**b**): novel management pathways for WEEE plastics. Reprinted with permission from ref. [92].

**Figure 33 polymers-14-05010-f033:**
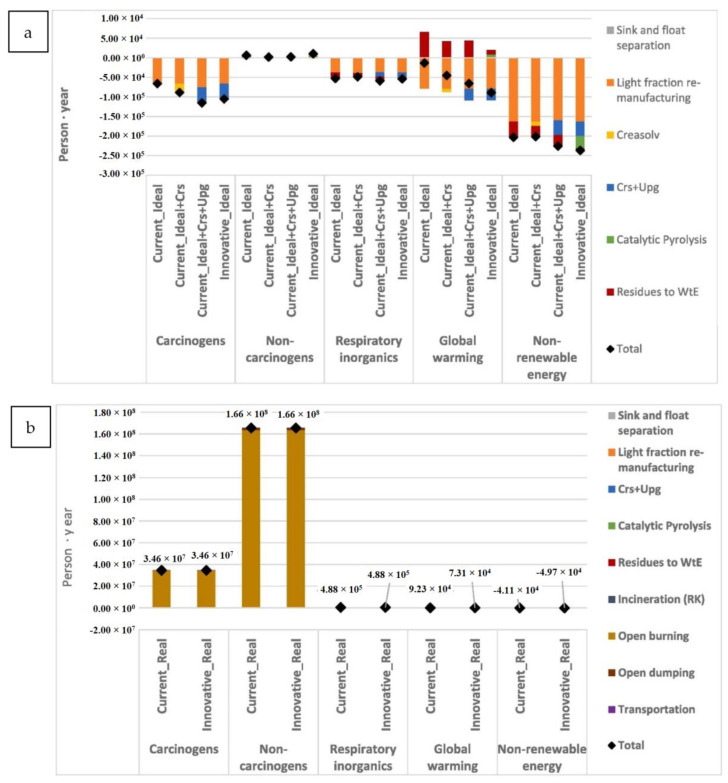
Normalized impact assessment data for the analyzed (**a**): Ideal and (**b**): Real scenarios concerning the functional unit and with the contribution of every single stage of the life cycle: Crs = CreaSolv^®^; Upg = Plastic Upgrading; Pyr = Catalytic Pyrolysis. The intermediates represent the current scheme where innovative processes are gradually added. The shaded rhombus indicates the total value for each impact category. Results are normalized in “Person · year”, i.e., the average impact in a specific category caused by a person during one year in Europe. Reprinted with permission from ref. [92].

**Figure 34 polymers-14-05010-f034:**
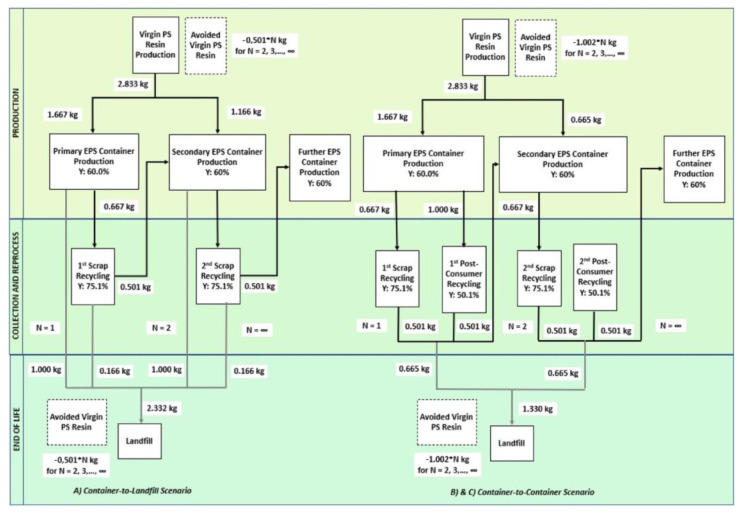
LCA flowchart used to assess the impact of EPS container production. Reprinted with permission from ref. [94].

**Figure 35 polymers-14-05010-f035:**
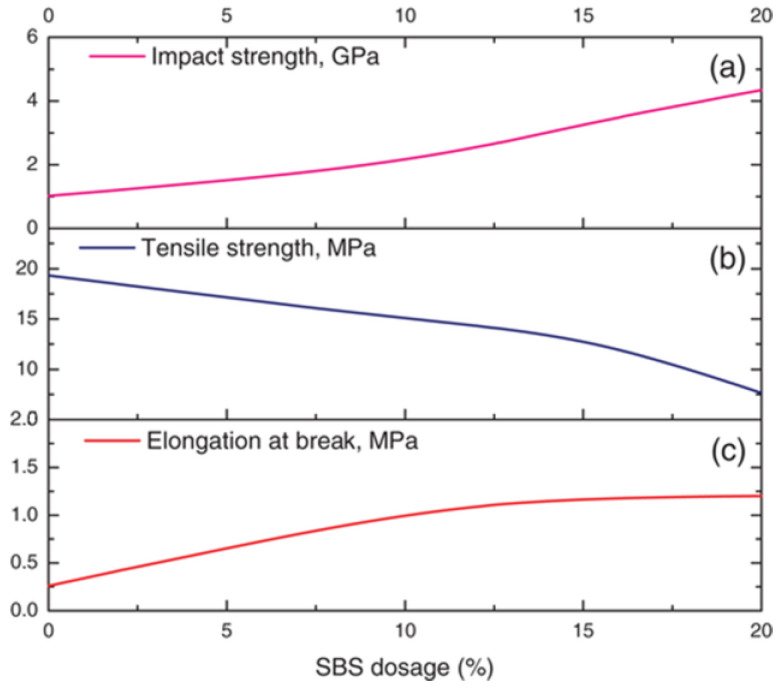
Variation in mechanical properties of WEEE composites with 0–20% SBS (**a**) Impact strength, (**b**) Tensile strength, (**c**) Elongation at break. Reprinted with permission from ref. [101].

**Figure 36 polymers-14-05010-f036:**
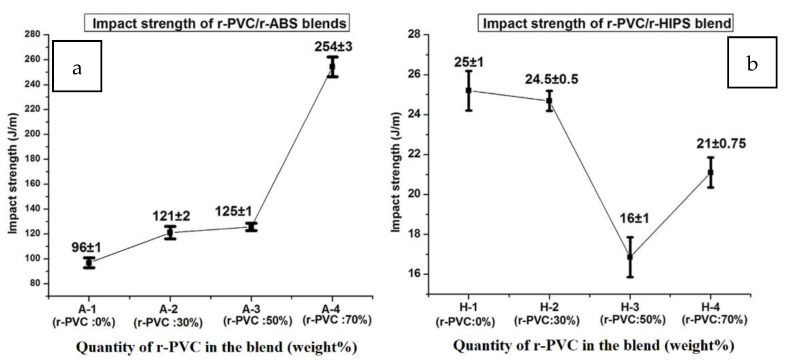
Impact strength of recycled PVC blended with (**a**) recycled ABS blends and (**b**) recycled HIPS. Reprinted with permission from ref. [102].

**Figure 37 polymers-14-05010-f037:**
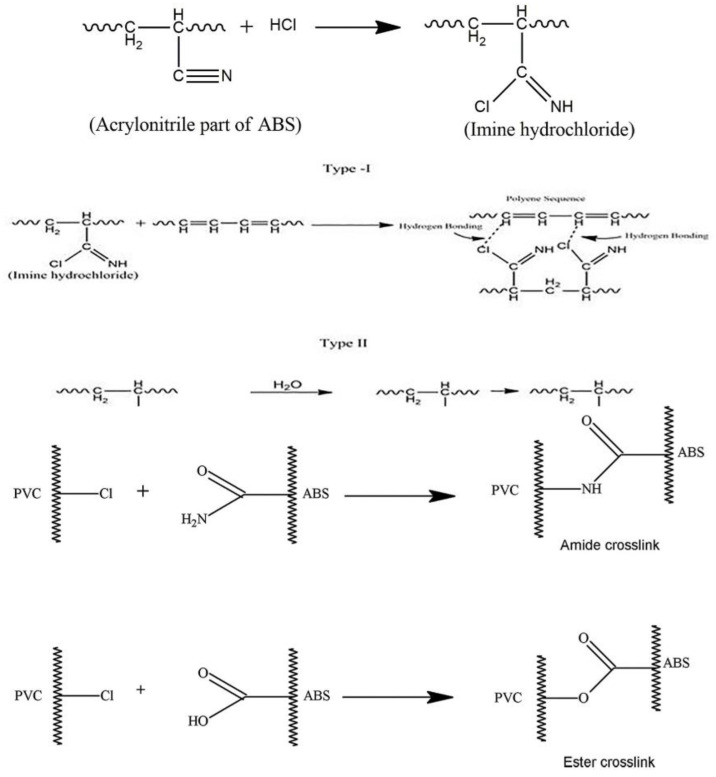
Schematic of the plausible cross-linking in recycled PVC/ABS blends. Reprinted with permission from ref. [102].

**Figure 38 polymers-14-05010-f038:**
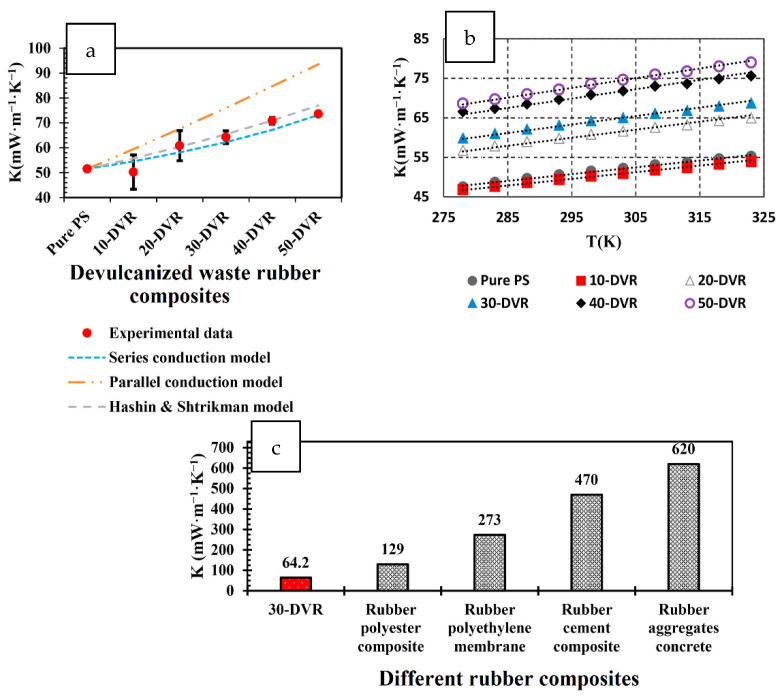
(**a**) Thermal conductivity of DVR–PS composites with varying quantities of DVR at 25 °C; (**b**) thermal conductivity of DVR–PS composite with varying temperatures; and (**c**) comparison of thermal conductivities of DVR–PS with other rubber composites. Reprinted with permission from ref. [103].

**Figure 39 polymers-14-05010-f039:**
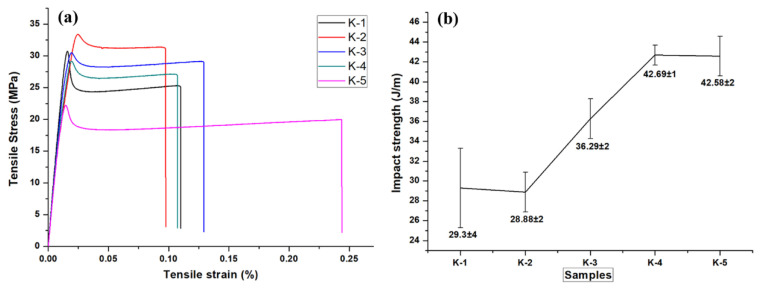
Mechanical properties (**a**): Tensile properties and (**b**): Impact strengths of recycled PS/HIPS/ABS ternary blends. Reprinted with permission from ref. [100].

**Figure 40 polymers-14-05010-f040:**
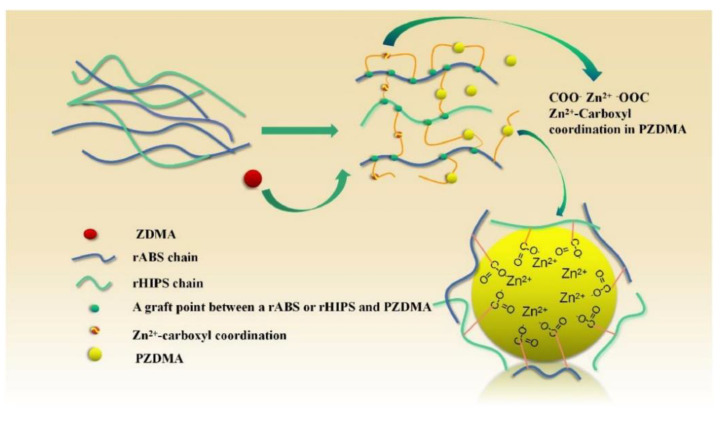
Mechanism of ionic cross-linking in recycled PS blends. Reprinted with permission from ref. [112].

**Figure 41 polymers-14-05010-f041:**
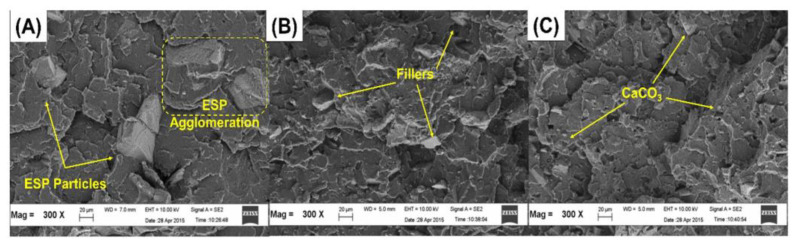
Evidence of homogenous CaCO_3_ dispersion in tensile fracture surfaces recycled PS/ESP/CaCO_3_ at (**A**) 80/20/0 wt.%, (**B**) 82/10/10 wt.%, and (**C**) 80/0/20 wt.%, respectively. Reprinted with permission from ref. [114].

**Table 1 polymers-14-05010-t001:** Specific signals of some mineral fillers and atmospheric species. Relative intensity correlates to the color, red (strong), orange (medium), yellow (weak), and light yellow (very weak). Adapted with permission from ref. [19].

Species	Wavenumber (cm^−1^)														
	1400	1300	1120	1100	1030	1015	910	885	870	790	760	750	710	690	670
Talcum				peak at 1010							761				670
Kaolin			1114		1032	1008	914			791		754		696	
CaCO_3_	peak at 1406								873				712		
TiO_2_								877		peak at 650, shoulder at 570–540					
CO_2_															670
H_2_O	Noise-like														

**Table 2 polymers-14-05010-t002:** Sample responses to single flame source tests. Adapted with permission from ref. [66].

Parameters	Type Samples								
	(1)	(2)	(3)	(4)	(5)	(6)	(7)	(8)	(9)
Does ignition occur?	Yes	Yes	Yes	Yes	Yes	Yes	Yes	Yes	Yes
Do flames reach 15 cm?	No	No	Yes	Yes	No	No	No	No	No
Time to reach 15 cm	-	-	5 s	2 s	-	-	-	-	-
Is the filter paper igniting?	No	No	No	No	No	No	No	No	No

**Table 3 polymers-14-05010-t003:** Mechanical and physicochemical properties of PS and sample composite. Adapted with permission from ref. [70].

Properties	Pure PS	Sample Composite
Density (Kg m^−3^)	600	300
Compression Strength (load in tons)	5.2	6.25
Water Absorption (%)	5	7.5
Thermal Conductivity (Wm^−1^ K^−1^)	0.033	0.029

**Table 4 polymers-14-05010-t004:** Domain sizes of polybutadiene (PB), styrene acrylonitrile (SAN), and polystyrene (PS) phase with different compatibilizers in HIPS/ABS blends. Adapted with permission from ref. [3].

Material	PB (μm^2^)	SAN (μm^2^)	PS (μm^2^)
HrAv-C_1_	0.14	5.2	-
HrAv-C_2_	0.15	3.6	-
HrAr-C_1_	0.37	11.33	-
HrAr-C_2_	0.34	22.00	-
ArHv-C_1_	0.21	-	4.2
ArHv-C_2_	0.13	-	4.0
ArHr-C_1_	0.11	-	3.7
ArHr-C_2_	0.11	-	7.2

Note: A = ABS; H = HIPS; v = virgin; r = recycled; C_1_ = SBS compatibilizer; and C_2_ = SEBS/SEB compatibilizer.

**Table 5 polymers-14-05010-t005:** Applications obtained for blend materials produced in [104] and the associated criteria and conformance standards. Adapted with permission from ref. [104].

Components Analyzed	Electrical Criterion	Mechanical Criterion	Application Standard	Polymers + GTR Suitable
Insulation for electric shepherds	Conductivity: <10^−12^ S cm^−1^ Tg δ < 10^4^	Tensile strength: 12.5 MPa Elongation at break: 300%	ITC-BT-39, 22, 23 24 UNE-EN 60335-2-76 IEC 60335-2-76	Eva + 10%
Spacer for power lines	Resistivity: >5.5 × 10^5^ Ω∙cm	Minimum tensile strength: 17.2 MPa Minimum elongation at break: 300%	IEC 61854	EVA + 20%
Universal electrical cable joint	Resistivity: >10^12^ Ω∙cm	Tensile strength: 12.5 MPa Elongation at break: 400%	IEC 60840 UNE HD 628	EVA + 10%
Filler for electrical applications	Resistivity: >10^12^ Ω∙cm	Tensile strength: 12.5 MPa Elongation at break: 350%	UNE 53 602; UNE 53 510; UNE-HD 632; UNE-EN 60811-4-1	EVA + 10%
Trays and pipes for electrical cables	Resistivity: >10^12^ Ω∙cm	Elongation at break: 80 ± 10% Tensile strength: 15 MPa	UNE EN 61537 UNE EN 50085-1: IEC 61537 (EN 61537)	PP + 10% EVA + 10%
Footwear for work use (insulating) Insulating: High electrical resistance	Resistivity: >10^6^ Ω∙cm >10^9^ Ω∙cm	Tensile strength: 10–12 MPa Elongation at break: >450%	UNE-EN ISO 20345/6/7:2005 UNE 53510	EVA + 10%

## Data Availability

Not applicable.

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
