# Peer review of "Upcycling Polystyrene"

_polymers, 2022, doi:10.3390/polym14225010_

Round 1

Reviewer 1 Report

Report on the manuscript

Upcycling Polystyrene

Jaworski C. Capricho , Krishnamurthy Prasad , Nishar Hameed , Mostafa Nikzad , Nisa Salim 

Manuscript ID: polymers-2001940

The study is a review regarding the problems arising from polystyrene waste. They focused on the loop that begins from the production process, waste generation till reuse. It presents the impact of the polystyrene waste and it’s recycling, as well as analyses the cycles of life for plastics.

Being a review the data presented has not that much impact but is well structured for the ones that are interested in the presented subject.

The manuscript is well written, the gathered data is well structured, presenting all the stages that involves polystyrene’s cycle of life as well as the proprieties that makes this product, in mixtures with others, having such a demand on the market.

Besides the chemical proprieties there are also presented the mechanical characteristics of the studied material, as well as structural images to get the idea of the influence of recycling on the structure.

I suggest publishing this paper in its current form, as it gathers enough data to emphasize the current research on polystyrene recycling and its advantages.

Author Response

Dear Reviewer,

We would like to thank you and the reviewers for your precious time and constructive comments and, most importantly, finding our research work important. 

Looking forward to hearing the outcome of the manuscript submission soon.

Sincerely,

Dr Nisa Salim

Reviewer 2 Report

·      Work entitled “Upcycling Polystyrene” is interesting. It discusses the different approaches for the recycling of polystyrene waste. However, the authors need to make some significant improvements in the manuscript.

According to the name of this review ‘‘Upcycling Polystyrene’’ the review must include a separate Figure of Polystyrene recycling methods, as in this work there are general plastic recycling methods related Figures. Take a glance on Figures 4, 5. In this way The review may be more specific than it is.

·       The full form of BASF should be mentioned.

·       Figures like, Figure number 1, 5, 6, 7, 10, 11, 15, 17, 19, 20, 21, 28, 30, 32, 34, 36, 37, 38, 39, and 41 should be of good resolution.

·       Pursing and Akin word may be replaced by following and similar respectively.

·       The color of all text is not same also sentence may be more effective by using easy words.

·       All Paragraphs are not in same format. All Paragraphs should be in same format.

·       In All Headings there must be same style of each starting Paragraph.

·       All headings should have same Writing Style. Check the writing style of POLYSTYRENE as heading 1.1 as it should be written as POLYSTYRENE.

·       The text requires significant attention in the correction of its grammatical, and typing mistakes. Look the words like analyses, tackling, teak, as these have spelling mistakes.

·       The sentences must not be too lengthy to understand it clearly.

·       Size of all figures is not equal and are not aligned properly. Check Figures like 9 (its sub parts), 15, 16, 17, 18, 20, 32, 38 and 41 for this, also Figure 13 is stretched. Align the remaining figures and their sub parts equally.

·       In Figure 17, the last part of the figure should be in the same line.

·       In Figure 25, Compressive strength MPa should be written at a proper distance from the axis.

·       REF in Figures may be written as Ref. to make it clearer.

·       The name of Figure 3 may be ‘‘Different forms of PS’’ rather than ‘‘Structure of PS’’.

·       The word syndiotactic (sPS) may be written as syndiotactic PS to mention that it is Syndiotactic Polystyrene.

·       In Table 1, look at the caption of table where it is written that Specific signals of some mineral fillers and atmospheric species. Relative intensity corre-284 lates to the color, with purple (very strong) as well as other colors are mentioned. But in table purple color in not shown.

·       In Table 3, in last Column sample composite should be written instead of Composite.

Points to highlight in Abstract and Conclusion:

·       The abstract and Conclusion should include the quantitative result framework obtained from the work.

·       A Graphical abstract may be added, to sum up, the total work and make it more effective.

·       The keywords in the abstract should include the word Polystyrene to show the main focus of the review.

·       In Conclusion, the word halt may be replaced by stop to understand it easily.

Points to highlight in Abbreviations:

·       All abbreviations are not mentioned at the end of the paper under heading Abbreviations. For example, OPS (Oriented polystyrene), Concrete foams (CFs), structural insulation panels (SIPs), grafted rubber concentrate (GRC), Styrene-Acrylonitrile Copolymers (SAN), acrylonitrile (AN), transparent impact polystyrene (TIPS), NIR-HSI (Near-Infrared Hyperspectral Imagery), FTIR-ATR (Fourier-Transform Infra-294 red Attenuated Total Reflection), poly(methyl methacrylate) (PMMA), chemical vapor deposition (CVD) process, graphene quantum dots (GQDs), design of experiments (DOE), carbon nanotubes (CNTs), poly(cyclo hexylethylene) (PCHE), acetone (AC) and ethyl acetate (EA), styrene monomer (SM), Contact angle (CA), lightweight concrete (LC), Bacterial cellulose (BC), greenhouse gas (GHG), thermogravimetry analysis (TGA), zinc dimethacrylate (ZDMA), Zn-salt poly(styrene-ran-cinnamic-acid) (SCA-Zn), styrene butadiene rubber (SBR), polyhedral oligomeric silsesquioxane (POSS), Ground tire rubber (GTR), International Electrotechnical Commis-1141 (IEC), etc.

·      1.     Writing style of headings 2.1, 2.2, 2.3…., 3.1, 3.2…. etc. should be same

2.     Heading no 3.3 Toughening Approaches of Recycled PS as an Upcycling Strategy should be numbered as 3 instead of 3.3.

3.     ‘Figure 9, 16, 33, 34 and 41’ should be bold (Word Figure) just like other figures.

4.     Alignment of all figures should be same (should be compatible with the text of the manuscript).

5.     Mention the x-axis of figure 9, 36 and 38.

6.     Mention the y-axis of figure 16 c and d, figure 18 and figure 20 c and d, figure 36.

7.     Mention the full-form of all abbreviations used in the manuscript.

8.     The manuscript discusses the three R’s: reduction, reuse, and recycling; it should either be reduction, reuse, recycle or reduction, reusing, recycling.

9.     In table no 1, relative intensity of ‘very strong’ mineral fillers is mentioned in the caption but is not highlighted in the table.

10.  The surface protection properties of hydrophobic organosilicon for use in waste PS-based lightweight concrete were analyzed in (sentence in incomplete).

11.  Such an analysis was conducted in ……..? where two types of a panel comprised of 15% and 30% recycled PS were 783 fabricated.

12.  Review the margin spacing of the subheadings (3.1, 3.2…..).

13.  At some points in the manuscript, abbreviations are discussed before the full-form is mentioned. Mention the full-form first then use the abbreviation.

In figure 2 (step propagation) review the mechanism, the generation of free radical is shown at wrong carbon. Below the carbon is mentioned with green arrow where the free radical has to be generated but it is mentioned above carbon with blue arrow this carbon has to be CH2 not CH.

Author Response

Dear Reviewer,

Sub: Response to the reviewers’ comment for the manuscript submitted to Polymers

Ref: Manuscript ID: polymers-2001940, titled, "Upcycling Polystyrene"

First of all, we thank you for allowing us to revise the manuscript. We would like to thank you and the reviewers for your precious time and constructive comments and, most importantly, finding our research work important. The changes and modifications have been highlighted in red in the revised manuscript.

We also take the opportunity to proofread the manuscript and correct a few typos.

We strongly believe that the quality of our manuscript is at a level that is acceptable for Polymers.

Looking forward to hearing the outcome of the manuscript submission soon.

Sincerely,

Dr Nisa Salim

Round 2

Reviewer 2 Report

No more comments, should be published